

# How to model crevasse initiation ? Lessons from the artificial drainage of a water-filled cavity on the Tête Rousse Glacier (Mont Blanc range, France)

Julien Brondex[1], Olivier Gagliardini[1], Adrien Gilbert[1], and Emmanuel Thibert[1]

[1]Univ. Grenoble Alpes, CNRS, IRD, Grenoble INP, INRAE, IGE, 38000 Grenoble, France

**Correspondence:** Julien Brondex (julien.brondex@univ-grenoble-alpes.fr)

**Abstract.** Crevasses play a crucial role in glacier-related hazards by facilitating water intrusion into the ice body and potentially triggering the collapse of large ice masses. However, the stress conditions governing their initiation and propagation remain uncertain. In particular, there is ongoing debate regarding the most relevant stress invariants to define fracture initiation (the failure criterion) and the corresponding failure strength, i.e. the stress threshold beyond which crevasses form. Laboratory estimates are hampered by the difficulty of reproducing natural glacier conditions, while in situ studies encounter uncertainties when converting strain or strain rate into stress estimates. This study investigates crevasse initiation processes by analyzing the artificial drainage of a water-filled cavity on Tête Rousse Glacier in 2010. Using the finite element code Elmer/Ice, we simulate the drainage and subsequent cavity refilling over three consecutive years. Given the well-constrained cavity geometry and water levels, stress fields are inferred directly from the force balance, removing the need to convert deformation data into stress estimates. Simulated stress distributions are compared with a pattern of circular crevasses mapped around the cavity after the first drainage event. Our results suggest that crevasse initiation is best explained by assuming a non-linear viscous mechanical response of ice (Glen's flow law, n = 3), rather than a linear viscous or linear elastic response. Additionally, by evaluating four failure criteria commonly used in glaciology, we show that the maximum principal stress criterion, with a stress threshold of 100 to 130 kPa, provides the best match to the observed crevasse field.

## 1 Introduction

Crevasses are essential components of the cryo-hydrologic system that have significant implications for glacier-related hazards due to their ability to: (i) channel water and the thermal energy it carries from the glacier surface to its bed (Irvine-Fynn et al., 2011; Chudley et al., 2021), with implication on basal friction and, consequently, glacier stability (Faillettaz et al., 2011); (ii) retain water, thereby increasing the risk of outburst floods (Bondesan and Francese, 2023); and (iii) serve as pre-existing fractures that may suddenly propagate through the ice, potentially triggering the collapse of large ice masses (Faillettaz et al., 2015; Chmiel et al., 2023). In addition, crevasses are involved in two processes that control the rate at which ice sheets lose mass into the ocean: iceberg calving (Colgan et al., 2016) and accelerated flow due to the softening of shear margins (Borstad et al., 2016). A proper representation of crevasses in numerical models is therefore a prerequisite for reliable forecasts of glacier instability on the one hand, and of the contribution of ice sheets to future sea level rise on the other hand.



Crevasses are fractures in the ice body that occur when internal stresses exceed a critical threshold marking the onset of failure (Colgan et al., 2016). Although stress fields are routinely computed in ice flow models, establishing a robust link between these stress fields and the occurrence of crevasses is not straightforward. This difficulty arises from the fact that, while failure is relatively well-defined under uniaxial stress, no universal failure theory exists for triaxial stress states. Collapsing a complex three-dimensional stress field into a single scalar equivalent stress, which can be compared to a stress threshold (or

failure strength) assumed to be an intrinsic and easily measurable material property, requires the definition of a failure criterion. To ensure independence from the choice of coordinate system, such criteria are typically formulated using combinations of the stress tensor invariants and its eigenvalues. Several failure criteria have been proposed in the literature to explain and model crevasse formation, but no clear consensus has emerged (e.g. Vaughan, 1993; Choi et al., 2018; Mercenier et al., 2019; Grinsted et al., 2024; Wells-Moran et al., 2025).

Furthermore, failure strength estimates from laboratory experiments show considerable discrepancies when compared to those derived from in situ observations. Laboratory estimates for tensile strength range from 700 to 3100 kPa for laboratory and lake ice (Petrovic, 2003), while field-based estimates typically fall within the range of 90 to 320 kPa (Vaughan, 1993; Grinsted et al., 2024; Wells-Moran et al., 2025), with the exception of Ultee et al. (2020), who reported a tensile strength on the order of 1000 kPa. Laboratory estimates are likely to be affected by the lack of representativeness of samples (e.g., sample

size, crystal fabric, grain size distribution, impurity content, and density) and of the applied stresses, which may not accurately reflect natural conditions (Vaughan, 1993; Petrovic, 2003). In contrast, studies based on in situ or remote sensing observations rely on converting strain rate measurements into stress estimates (Vaughan, 1993; Grinsted et al., 2024; Wells-Moran et al., 2025). This process introduces significant uncertainties due to: (i) the need for assumptions regarding ice rheology; (ii) spatial and temporal averaging of velocities used to produce the strain rate field; (iii) the limitation of observations to the surface; and

(iv) the fact that, when advection is significant, the local stress field at a crevasse location may not reflect the conditions under which it was formed.

Because they constitute full-scale experiments characterized by a sudden evolution of stresses which causes the formation of circular crevasses, ice subsidence events have been used to advance knowledge of ice fracture processes (e.g., Evatt and Fowler, 2007; Ultee et al., 2020). For instance, Ultee et al. (2020) analyzed the 2015 eastern Skaftá cauldron collapse (Vatnajökull ice

cap, Iceland) following the sudden drainage of a large subglacial lake to estimate the tensile strength of ice. However, due to missing data regarding the exact geometry of the cavity, such studies typically rely on surface deformation observations to infer stresses, or on simplified models that approximate the poorly constrained three-dimensional (3D) geometry of the problem as, e.g., a 2D viscous beam (Evatt and Fowler, 2007) or an idealized circular viscoelastic plate (Ultee et al., 2020). Furthermore, the transient nature of the drainage is generally disregarded as data on the temporal evolution of water level in the cavity is

usually unavailable or poorly constrained.

In this study, we use the carefully monitored artificial drainage of a water-filled cavity on Tête Rousse Glacier (Mont Blanc range, France) to constrain the failure criterion and stress threshold that best reproduce the field of circular crevasses mapped at the surface during the summer following the pumping operation. To this end, we performed numerical simulations using the finite element code Elmer/Ice. Since the evolution of the water level and the cavity volume (and, to a lesser extent, its geometry)



are well-constrained, water pressure against the cavity wall can be prescribed as a time-dependent boundary condition, and stress fields are inferred directly from the force balance. In other words, our numerical experiments are entirely force-driven. The resulting deformations serve as independent control variables to assess the nature of the mechanical response, which stands in sharp contrast to studies that infer stress fields from observed displacements or velocities.

The paper is organized as follows: in Sect. 2, we present the study site and the available data.; in Sect. 3, we introduce the
model and describe the numerical experiments; results are presented in Sect. 4 and discussed in Sect. 5 in light of previous work.

## 2 Study Site and Data

### 2.1 Study Site and historical background

Tête Rousse Glacier is a small glacier of the Mont Blanc massif (French Alps, $45°55'$ N, $6°57'$ E). It is approximately 200 m
wide and 500 m long in the east–west direction, covering a total surface area estimated at $0.08$ km$^2$ as of 2007 (Fig. 1a). Its surface topography ranges from 3110 m above sea level (a.s.l) to 3260 m a.s.l. This glacier would likely have remained largely unnoticed among the more prominent glaciers of the surroundings, were it not for the catastrophic event of July 1892, when the collapse of a water pocket drained $\sim$200,000 m$^3$ of water and ice, claiming 175 lives in the village of Saint-Gervais-Le Fayet. Between 2007 and 2010, extensive geophysical surveys revealed the presence of a new subglacial cavity containing
an estimated water volume of $53\,500$ m$^3$. In two boreholes drilled above the cavity, artesian outflows rising 10-20 cm above the glacier surface were observed, indicating that the water pressure exceeded the ice overburden pressure. In light of these findings, public authorities ordered the drainage of the cavity, which was carried out between 25 August and 8 October 2010 using down-hole pumps. A total of $47\,700$ m$^3$ of water was extracted from the glacier, leaving a few thousands cubic meters that could not be pumped out. In August 2011, circular crevasses were detected at the glacier surface around the cavity (Fig.
1b). The cavity naturally refilled during summer 2011 and needed to be drained again in September of the same year. The same thing happened in summer 2012. Meanwhile, the cavity's geometry evolved towards a reduced volume, due to ice creep during periods of low water levels. Several detachments of ice blocks from the cavity roof were also observed on the field. Volume loss was further accelerated by the partial collapse of the cavity roof on 14 August 2012 (Fig. 1c), which likely caused the refreezing of significant volumes of water in the pores between the ice block debris. Consequently, the cavity's volume decreased from
an estimated $53\,500$ m$^3$ in 2010 to $12\,750$ m$^3$ in 2013. This volume reduction, combined with the control of water pressure through numerous holes drilled from the surface, led to the decision to cease further drainage operations after 2013. Since the water-filled cavity was discovered in 2010, the Tête Rousse Glacier has been meticulously monitored, leading to the collection of extensive data and the publication of numerous studies (Vincent et al., 2010; Legchenko et al., 2011; Gagliardini et al., 2011; Vincent et al., 2012; Gilbert et al., 2012; Legchenko et al., 2014; Vincent et al., 2015; Garambois et al., 2016).



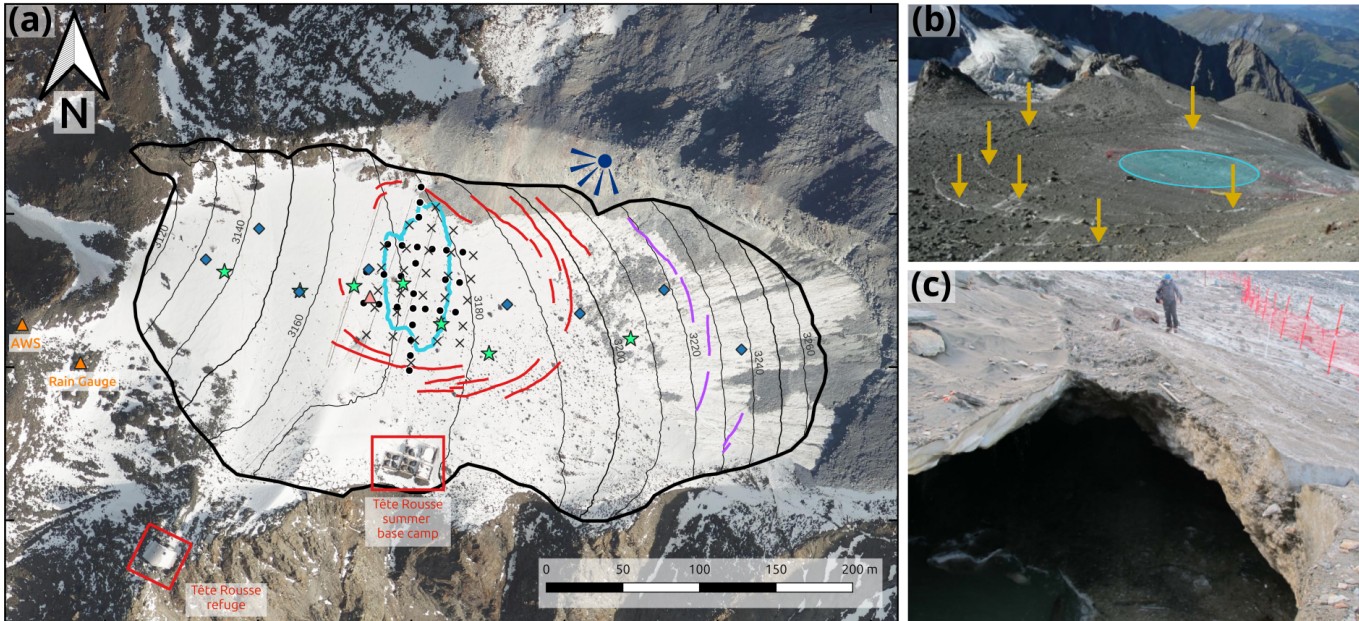

**Figure 1. (a)** Elevation contours of Tête Rousse Glacier as of 2011 overlaid on an aerial orthophoto (IGN) from 30 September 2023; the thick black line outlines the modeled domain; the cyan line marks the cavity outline; red lines indicate circular crevasses mapped in summer 2011, whereas magenta lines correspond to crevasses unrelated to the cavity drainage; black dots and crosses represent the stake networks used to survey displacements in 2010 and 2011, respectively; blue diamonds denote stakes used for surface mass balance measurements; triangles indicate meteorological stations: the Automatic Weather Station and rain gauge (orange triangles) and the Surface Energy Balance station (pink triangle); green stars mark the borehole locations used for the 2010 temperature measurements; the blue pictogram on the northern moraine indicates the approximate position from which photo (b) was taken. **(b)** Photograph of the glacier taken in summer 2011, showing circular crevasses (highlighted by yellow arrows) surrounding the cavity roof (approximately outlined by the cyan circle). **c** Photograph taken at the end of summer 2012, after the collapse of a small part of the cavity roof.

## 2.2 Data

Below, we provide a brief overview of the data used in this study. We refer the reader to the literature introduced above for detailed information on data acquisition.

### 2.2.1 Surface, bedrock and subglacial cavity topographies

The surface Digital Elevation Model (DEM) of the glacier was obtained from laser-scan measurements conducted over the entire glacier surface on 10 August 2011 (Vincent et al., 2015). The bedrock DEM was derived from several Ground Penetrating Radar (GPR) surveys (Garambois et al., 2016). The cavity topography was reconstructed as follows: (1) an initial estimate of the cavity geometry was derived from sonar measurements performed in September 2010 and September 2011 (Vincent et al., 2015), supplemented by GPR surveys conducted in May 2010, May 2011 and August 2011 (Garambois et al., 2016); (2)



manual adjustments were applied to achieve an initial volume of approximately $51\,000\ \mathrm{m}^3$, which falls between the $53\,500\ \mathrm{m}^3$ of water estimated from surface nuclear magnetic resonance (SNMR) measurements performed in September 2009 and June 2010 and the $47\,700\ \mathrm{m}^3$ pumped out of the cavity in 2010 (Legchenko et al., 2014). As a result, the initial position of the cavity within the glacier and its volume are well constrained, but its exact shape remains more uncertain.

### 2.2.2 Surface mass balance

Between 2010 and 2013, annual and seasonal mass balances were measured at eight stakes on Tête Rousse Glacier (blue diamonds, Fig. 1a). These measurements were used to calibrate a degree-day model which, combined with meteorological data acquired both in situ (triangles, Fig. 1a) and at the Chamonix station, enabled the reconstruction of daily Surface Mass Balance (SMB) across the glacier. More details on these measurements in Vincent et al. (2015).

### 2.2.3 Ice temperature

Seven boreholes were drilled using hot water drilling in July 2010 along a central longitudinal section of the glacier (Fig. 1a). In each borehole, temperature was measured throughout the entire ice thickness using thermistor chains with an accuracy of $\pm 0.1$ C (Gilbert et al., 2012). A reference temperature field was constructed from these measurements, assuming no variations of temperature in the transverse (south-north) direction. Specifically, a 2D vertical temperature field was generated by projecting temperatures measured at all thermistors onto a common west-east longitudinal cross-section. Linear interpolation was applied between measurement points, while simulation results from Gilbert et al. (2012) were used to extrapolate temperatures in regions beyond the interpolation grid. The resulting 2D vertical temperature field is shown in Fig. S1 of the Supplement.

### 2.2.4 Water-level measurements

The evolution of the mean water level within the cavity over time was reconstructed from piezometer measurements described in Vincent et al. (2015). Short-term fluctuations were smoothed, and data gaps were filled using linear interpolations (see Fig. 2a in Vincent et al. (2015)). The resulting water level curve, used as a forcing for our prognostic simulations, is shown in Fig. 2.

### 2.2.5 Topographic measurements

The displacements in the vicinity of the cavity roof in response to water-level fluctuations was surveyed using a network of 27 stakes in 2010 (black dots in Fig. 1a) and 29 stakes in 2011 (black crosses in Fig. 1a). Several stakes were lost over the season. Stake positions were measured using a total station with an accuracy of $\pm 0.005$ m (Vincent et al., 2015).

### 2.2.6 Crevasse positions

Circular crevasses were observed around the cavity roof in August 2011 after the snow had melted away (yellow arrows Fig. 1b). These crevasses were mapped using a differential GPS on the 30 and 31 August 2011 (red lines, Fig. 1a). Two of these





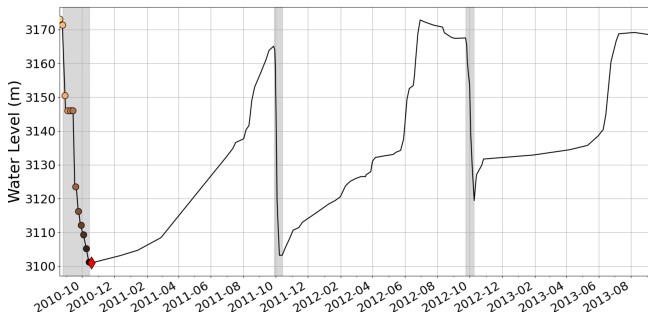

**Figure 2.** Reconstructed water level inside the cavity from the beginning (20 August 2010) to the end (5 September 2013) of the prognostic simulations. Grey shaded areas correspond to pumping operations. Colored dots correspond to days for which equivalent stress profiles are plotted in Fig. 6. The red diamond corresponds to the day for which maps of equivalent stress and pressure are given in Figs. 5 and 9, respectively.

crevasses, one downstream and one upstream of the cavity, were instrumented with stakes placed opposite each other to monitor their opening and closing. Measurements taken between 9 September and 21 October 2011 confirmed that their behavior was

governed by water level fluctuations within the cavity. Specifically, when the cavity was pressurized (9–23 September 2011), the upstream crevasse closed at a rate of up to 5 mm/day, whereas after drainage (starting 23 September 2011), it reopened at a rate of up to 3 mm/day. A few additional crevasses were mapped much further upstream of the glacier (magenta lines, Fig. 1a), but they are unlikely to be related to cavity drainage. Although the crevasses were discovered late in the season due to prolonged snow cover, we are confident that they were mapped within less than one meter of their original formation sites,

given the very low surface horizontal velocities ($< 1 \mathrm{~m~a^{-1}}$).

## 3    Methods

In this Section, we give an overview of the model, introduce the various failure criterion investigated, and describe the experimental setup. The full set of model equations is presented in Appendix A, while variables, parameters, and constants used in our model are summarized in Table 1.

### 3.1    Model description

In this study, our primary goal is to establish a correlation between the occurrence of circular crevasses and the distribution and magnitude of stresses. We are thus interested in modeling the evolution of the stress and displacement fields in response to fluctuations of water pressure within the cavity. This requires solving the momentum balance equation together with a constitutive law, which relates stresses to strain and/or strain rates. Any load applied to polycrystalline ice will result in an instantaneous

elastic deformation accompanied by a time-dependent creep (viscous) deformation. Over short timescales, glacier ice is thus usually viewed as a compressible isotropic linear-elastic solid. Beyond the so-called Maxwell timescale (see Sect. 5.1), viscous





deformations largely dominate over elasticity and ice is best described as an incompressible non-linear viscous fluid. Some authors have adopted viscoelastic models to capture both regimes within a unified framework (e.g., Christmann et al., 2019; Ultee et al., 2020; Hageman et al., 2024). By contrast, following Mosbeux et al. (2020), we model both regimes separately. We argue that this approach is justified, as the elastic response is largely negligible for our case study, as we will demonstrate in this paper. For both framework, we assume ice is isotropic.

The linear elastic regime of isotropic ice is characterized by a linear increase of strain with applied stress according to Hooke's law. The key parameters of Hooke's law are Young's modulus $E$ and Poisson's coefficient $\nu$. Experimental studies investigating the propagation of sound waves in ice have reported Young's modulus values of about $9\,\mathrm{GPa}$ (Cuffey and Paterson, 2010). Authors who have used simple elastic beam or plate models to reproduce the observed flexure of ice shelves/ice sheet margins in response to ocean tides have inferred lower values, e.g., $E = 0.88 \pm 0.35\,\mathrm{GPa}$ for Vaughan (1995), $E = 0.8 - 3.5\,\mathrm{GPa}$ for Schmeltz et al. (2002). In their study, Ultee et al. (2020) tested the range $E = 0.1 - 10\,\mathrm{GPa}$, with $E = 1\,\mathrm{GPa}$ being reported as the representative value. In contrast, the standard value for Poisson's ratio, $\nu = 0.3$, is more widely accepted among authors (Cuffey and Paterson, 2010). It is important to note that for any $\nu < 0.5$, ice is implicitly treated as a compressible material. In this case, strict mass conservation would require updating the density to account for volume changes. However, these deformations are assumed to be sufficiently small that density can be considered constant. The resulting system of equations is solved for the three components of the displacement field. The domain geometry evolves instantaneously in response to changes in external forces and is directly inferred from the displacement field.

The viscous behavior of ice is described by Glen's flow law, a Norton-Hoff-type relationship that, assuming ice incompressibility, relates deviatoric stresses to strain rates through a stress-dependent viscosity, making it inherently non-linear. Viscosity is highly sensitive to temperature. Furthermore, Continuum Damage Mechanics (CDM) can be used to account for the degradation of ice mechanical properties at the mesoscale caused by the nucleation of cracks that are too small to be explicitly resolved in the continuum model. This is achieved by prescribing a dependence of viscosity on a physical state variable known as damage (e.g., Krug et al., 2014). Damage initiates in regions where the prescribed failure criterion is met, is advected with the ice flow, and feeds back on ice viscosity (see Appendix A). In this work, we use the damage model implemented within Elmer/Ice by Krug et al. (2014). Unlike the elastic response, where deformation is instantaneous, the viscous regime is characterized by a transient evolution of the geometry toward a new steady state following a change in external forces, with the stress and deformation fields evolving in a coupled manner due to the non-linear rheology of ice. The geometry evolution is incorporated into the model by solving a free surface equation on the upper surface, which accounts for flow divergence and surface mass balance. To capture the dynamic evolution of the subglacial cavity in response to variations in water pressure, the lower surface of the glacier is also treated as a free surface (neglecting accretion/melting) with a lower bound given by the bedrock elevation. Concretely, the lower surface of the glacier is either in contact with the bedrock or forms the walls of the cavity. When the water level in the cavity rises, previously grounded ice may begin to float. Conversely, if the water level drops, floating ice can re-ground. This contact problem is analogous to the dynamics of a marine ice sheet's grounding line, the boundary where grounded ice begins to float and forms an ice shelf. To accurately simulate the evolution of the cavity geometry in transient





**Table 1.** List of variables, parameters and constants used in our model.

| Symbol | Name | Equation/Value(s) | Unit |
|---|---|---|---|
| Parameters | | | |
| $B$ | Damage enhancement factor | Table 2 | − |
| $C_\mathrm{W}$ | Weertman friction coefficient | 0.1 | MPa a m$^{-1}$ |
| $E$ | Young's modulus | 1 and 9 | GPa |
| $n$ | Glen's law exponent | 3 | − |
| $\alpha$ | First coefficient of Hayurst eq. stress | 0.21 | − |
| $\beta$ | Second coefficient of Hayurst eq. stress | 0.63 | − |
| $\mu$ | Friction coefficient of Coulomb eq. stress | 0.1 | − |
| $\nu$ | Poisson's ratio | 0.3 | − |
| $\sigma_\mathrm{th}$ | Stress threshold | Table 2 | kPa |
| Constants | | | |
| $A_0$ | Reference fluidity for $T > -10$ C | $2.43 \times 10^{-2}$ | Pa$^{-3}$s$^{-1}$ |
| $g$ | Gravitational constant | 9.81 | m s$^{-2}$ |
| $Q$ | Activation energy for $T > -10$ C | 115 | kJ mol$^{-1}$ |
| $R$ | Gas constant | 8.314 | J mol$^{-1}$ K$^{-1}$ |
| $\rho_\mathrm{i}$ | Ice density | 917 | kg m$^{-3}$ |
| $\rho_\mathrm{w}$ | Water density | 1000 | kg m$^{-3}$ |

simulations, we employ dedicated libraries developed within Elmer/Ice that have been specifically designed to address this type of problem (Durand et al., 2009).

The computational domain is bounded by three surfaces: the glacier upper surface, the lateral boundary, and the glacier lower surface. The upper surface is treated as a stress-free boundary. At the lateral boundary, a non-penetration condition is enforced in the normal direction, while a free-slip condition applies in the tangential direction. The lower surface of the glacier is subject to two distinct boundary conditions, depending on whether the ice is in contact with the bedrock or forms part of the cavity walls. Within the cavity, the measured water level is used to impose hydrostatic water pressure while a no-displacement (resp. no-velocity) condition is applied elsewhere. Given the very low observed surface velocities and the presence of cold ice in the lower part of the glacier, we anticipate little to no sliding at the ice/bedrock interface (Vincent et al., 2015). For transient simulations within the viscous framework, solving the contact problem using Elmer/Ice libraries requires prescribing a sliding law (Durand et al., 2009). As a consequence, we implement a linear Weertman law with a very strong friction coefficient. The resulting sliding velocities are systematically much lower than $1$ m a$^{-1}$.



## 3.2 Failure criteria

We evaluate four of the most frequently used failure criteria in glaciology-related studies, which are introduced below. We denote the three Cauchy principal stresses as $\sigma_\mathrm{I}$, $\sigma_\mathrm{II}$, and $\sigma_\mathrm{III}$, adopting the convention $\sigma_\mathrm{III} < \sigma_\mathrm{II} < \sigma_\mathrm{I}$.

### 3.2.1 Coulomb criterion

The Coulomb criterion, initially formulated to describe shear failure in rocks, has since been adapted to ice (MacAyeal et al., 1986; Vaughan, 1993; Weiss and Schulson, 2009; Wells-Moran et al., 2025). The Coulomb equivalent stress is expressed as:

$$\sigma_\mathrm{eq,C} = \frac{\sigma_\mathrm{I} - Ice\sigma_\mathrm{III}}{2} + \mu\frac{\sigma_\mathrm{I} + \sigma_\mathrm{III}}{2}, \tag{1}$$

where $\mu$ denotes the internal friction coefficient, set here to $\mu = 0.1$, following MacAyeal et al. (1986) and Vaughan (1993). The first term on the right-hand side represents the Tresca criterion, widely applied in engineering, which is symmetric under tension and compression and does not depend on pressure. The second term accounts for frictional resistance along fault planes, increasing compressive strength relative to tensile strength, and introducing a pressure dependence to the failure criterion.

### 3.2.2 von Mises criterion

The von Mises (vM) criterion is an energy-based criterion frequently used to model calving (e.g., Morlighem et al., 2016; Choi et al., 2018; Mercenier et al., 2019) and to explain or model crevasse opening (e.g., Vaughan, 1993; Albrecht and Levermann, 2012, 2014; Grinsted et al., 2024; Wells-Moran et al., 2025). Initially proposed within plasticity theory to predict the yielding of ductile materials, this criterion states that yielding begins when the elastic energy of distortion (i.e., the deviatoric component of deformation energy) reaches a critical value. Expressed solely in terms of deviatoric stresses, the von Mises criterion remains independent of isotropic pressure and is symmetric in tension and compression. The von Mises equivalent stress reads:

$$\sigma_\mathrm{eq,vM} = \sqrt{\frac{1}{2}\left[\left(\sigma_\mathrm{I} - \sigma_\mathrm{II}\right)^2 + \left(\sigma_\mathrm{I} - \sigma_\mathrm{III}\right)^2 + \left(\sigma_\mathrm{II} - \sigma_\mathrm{III}\right)^2\right]}, \tag{2}$$

where deviatoric principal stresses have been rewritten in terms of Cauchy principal stresses.

### 3.2.3 Maximum principal stress criterion

The Maximum Principal Stress (MPS) criterion, also known as Rankine criterion, defines the equivalent stress solely based on the highest eigen value of the Cauchy stress tensor, $\sigma_\mathrm{I}$. Notably, $\sigma_\mathrm{I}$ can be either tensile or compressive. However, most studies applying this criterion to glacier and ice sheet damage consider only tensile stresses (e.g., Krug et al., 2014; Mercenier et al., 2018, 2019). In such a case, the equivalent stress reads:

$$\sigma_\mathrm{eq,MPS} = \max\left(0, \sigma_\mathrm{I}\right). \tag{3}$$

This criterion is pressure-dependent as the Cauchy principal stresses include the effect of pressure.



### 3.2.4 Hayurst criterion

The criterion was initially proposed by Hayhurst (1972) to describe the creep rupture of metallic alloys at a fraction of their yield stress when tested at temperatures between $40$ and $60$ % of their melting point. It is arguably the most widely used criterion in glaciology-related applications of Continuous Damage Mechanics (e.g., Pralong and Funk, 2005; Jouvet et al., 2011; Duddu and Waisman, 2012; Duddu and Waisman, 2013; Mobasher et al., 2016; Jiménez et al., 2017; Mercenier et al., 2018, 2019; Huth et al., 2021; Huth et al., 2023; Ranganathan et al., 2024). The Hayhurst equivalent stress is given by:

$$\sigma_{\mathrm{eq,H}} = \alpha\sigma_{\mathrm{I}} + \beta\sigma_{\mathrm{eq,vM}} + (1 - \alpha - \beta)\,\mathrm{tr}(\boldsymbol{\sigma}), \tag{4}$$

indicating that the Hayhurst criterion is a linear combination of the maximum principal stress criterion, the von Mises criterion, and the isotropic pressure $p = -\mathrm{tr}(\boldsymbol{\sigma})/3$. The parameters $\alpha$ and $\beta$ control the relative contributions of these terms and must satisfy:

$$0 \le \alpha, \beta, (1 - \alpha - \beta) \le 1. \tag{5}$$

Here, we follow Pralong and Funk (2005) and take $\alpha = 0.21$ and $\beta = 0.63$.

## 3.3 Experimental setup

Below, we present the numerical setup and describe the two types of experiments, diagnostic and prognostic, on which this study relies.

### 3.3.1 Numerical setup

All equations of the model are solved using the open-source finite-element code Elmer (https://github.com/ElmerCSC/elmerfem) and its glaciological extension, Elmer/Ice (Gagliardini et al., 2013). Two initial computational domains are constructed from the dataset introduced in Sect. 2.2.1 : one including the subglacial cavity and one without any cavity. In both cases, the upper glacier surface is extracted from the 2011 surface DEM. For the domain without a cavity, the lower glacier surface is directly extracted from the 2014 bedrock DEM. For the domain including the cavity, the reconstructed cavity topography is used to initialize the bottom free surface in the region where the ice is not in contact with the bed. For both computational domains, the finite-element mesh is generated by vertically extruding a 2D horizontal footprint comprising 5842 linear triangles over 16 vertical layers, from the lower to the upper glacier surface. To ensure comparable meshes, the horizontal footprint is refined around the projected cavity contour even in the cavity-free case, leading to a typical node spacing of 2 m within the expected cavity area, increasing to 12 m at the glacier's lateral boundaries. As a result, the final mesh consists of 53,727 nodes (93,472 elements) for both computational domains. To optimize CPU time, the mesh is divided into 8 partitions for parallel computing.

### 3.3.2 Diagnostic experiments

A first set of diagnostic experiments is conducted to assess the signature of an empty cavity on the surface stress field. This approach serves as a means of quantifying the immediate response of the stress field to the instantaneous drainage of a cavity





250  initially in pressure equilibrium, which we consider to be an unrealistic worst case scenario in terms of stress shock. All
simulations are run in pairs: each simulation is first conducted on the cavity-free domain and then repeated on the domain
containing a cavity. In the latter case, the cavity is assumed to be empty, meaning that water pressure is set to zero throughout
the cavity. Both the elastic and viscous frameworks are considered. For the elastic framework, we test two end-member values
of Young's modulus that bound the typically accepted range: $E = 1$ GPa and $E = 9$ GPa. We argue that using lower Young's

255  modulus values to fit observations in a purely elastic model may overlook the viscous component of the observed deformation
(see Sect. 5.1). For the viscous framework, we consider first the usual non-linear Glen's flow law, i.e. $n = 3$ in Eq. (A5). A
first pair of simulations is conducted using the reference temperature field introduced in Sect. 2.2.3. To assess how temperature
influences the results, the pair of diagnostic simulations is repeated twice with uniform ice temperatures of $T = 0$ C and
$T = -2$ C, respectively. In addition, we consider the case of a linear viscous law, i.e. $n = 1$ in Eq. (A5), instead of the non-

260  linear Glen's law. Since the parameters of the Arrhenius law (A8) were derived for $n = 3$ (Cuffey and Paterson, 2010), adopting
$n = 1$ requires an adjustment of the fluidity. We test two uniform fluidity values: $A = 0.4\,\mathrm{MPa^{-1}a^{-1}}$ and $A = 0.8\,\mathrm{MPa^{-1}a^{-1}}$.
These values are chosen so that the computed velocities remain of the same order as those obtained using the non-linear Glen's
flow law.

A separate set of diagnostic experiments is carried out within the elastic framework to quantify the expected elastic defor-

265  mation associated with the drainage of the water in the cavity between two specific dates. Pairs of simulations are run on the
computational domain with cavity, applying water pressure corresponding to the reconstructed water level at considered pair
of dates. The displacement field computed for the later date is then subtracted from that computed for the earlier one. While
these simulations neglect the viscous evolution of the geometry over time, the resulting displacement field offers an estimate
of the order of magnitude of the total elastic deformations expected between the two dates.

### 3.3.3  Prognostic experiments

To simulate the evolution of the stress field and deformations over time in response to cavity water pressure changes, transient
simulations are performed within the viscous framework using the non-linear Glen's law with $n = 3$ only. These simulations
run continuously from 20 August 2010 to 5 September 2013, covering the three drainage sequences of 2010 (26 August to
15 October), 2011 (28 September to 14 October), and 2012 (23 September to 9 October), along with the refilling phases

275  in between (Fig. 2). The simulations start six days before the first pumping sequence to allow for a short relaxation period,
reducing deformation anomalies linked to the initial model configuration. These anomalies notably arise from the upper surface
being representative of August 2011, while the initial water level and cavity volume correspond to that of August 2010, along
with uncertainties in the exact cavity shape. Since the cavity was out of pressure equilibrium when discovered, as evidenced
by artesian outflow observed in two boreholes, a longer spin-up period would have led to cavity expansion and downstream

280  migration, resulting in an initial geometry that would deviate too much from observations. All pumping sequences are simulated
with a one-day timestep, while the periods in between are simulated with a five-day timestep. Water pressure is applied in the
cavity, i.e. at nodes identified as being above the bedrock. This pressure is derived from the reconstructed water level shown in
Fig. 2, following Eq. (A16). The reconstructed daily SMB introduced in Sect 2.2.2 is applied across the glacier surface.





**Table 2.** Combinations of parameters $B$ and $\sigma_{\text{th}}$ tested.

| Combi. N° | $B$ [$-$] | $\sigma_{\text{th}}$ [kPa] |
|:---:|:---:|:---:|
| 1 | 1.34 | 130 |
| 2 | 1.16 | 80 |
| 3 | 0.97 | 60 |
| 4 | 0.78 | 190 |
| 5 | 1.91 | 120 |
| 6 | 0.59 | 150 |
| 7 | 1.72 | 100 |
| 8 | 1.53 | 170 |

A reference simulation is performed without accounting for damage in order to establish a correlation between the stress
state and the initiation of crevasses in intact (i.e., undamaged) ice. This simulation is run using the temperature field introduced
above. It is then repeated replacing the spatially-distributed temperature field with uniform temperatures of $T = 0$ C and
$T = -2$ C. Following this, a set of simulations based on the reference simulation is run with the damage model enabled,
considering each of the four failure criterion introduced above alternatively. These simulations include the feedback of damage
on viscosity according to Eqs. (A7)-(A9). Since the amount of damage created depends critically on the values assigned to
the damage enhancement factor $B$ and stress threshold $\sigma_{\text{th}}$ according to Eqs. (A10) - (A11), we test eight combinations of
these parameters for each failure criterion. The combinations are generated using a Latin Hypercube Sampling (LHS), with
$\sigma_{\text{th}}$ spanning the range 50-200 kPa and $B$ spanning the range 0.5-2.0 (Krug et al., 2014). The resulting values of $B$ and $\sigma_{\text{th}}$
are summarized in Table 2.

## 4 Results

### 4.1 Signature of the cavity on diagnostic surface stresses

Figure 3 shows the signature of an empty cavity in terms of the diagnostic (instantaneous) maximum principal stress com-
puted at the surface of the glacier for the three constitutive laws considered. Results are presented for $E = 1$ GPa, $A = 0.4$ MPa$^{-1}$ a$^{-1}$, and the reference temperature for the linear elastic, linear viscous, and non-linear Glen's laws, respectively.
As shown in Figs S2-S4 of the Supplement, the computed diagnostic stress field is independent of the law parameters within
each of the three constitutive laws; only the resulting displacements/velocities are affected. Specifically, Young's modulus for
the linear elastic law and fluidity for the linear viscous law act as scaling factors for displacements and velocities, respectively.
For example, displacements (resp. velocities) computed with $E = 1$ GPa (resp. $A = 0.8$ MPa$^{-1}$ a$^{-1}$) are exactly 9 times (resp.
twice) those computed with $E = 9$ GPa (resp. $A = 0.4$ MPa$^{-1}$ a$^{-1}$). The same applies to the non-linear Glen's law, where
different temperature fields lead to different velocity fields, while the stress field remains unaffected. As discussed in Section 5,




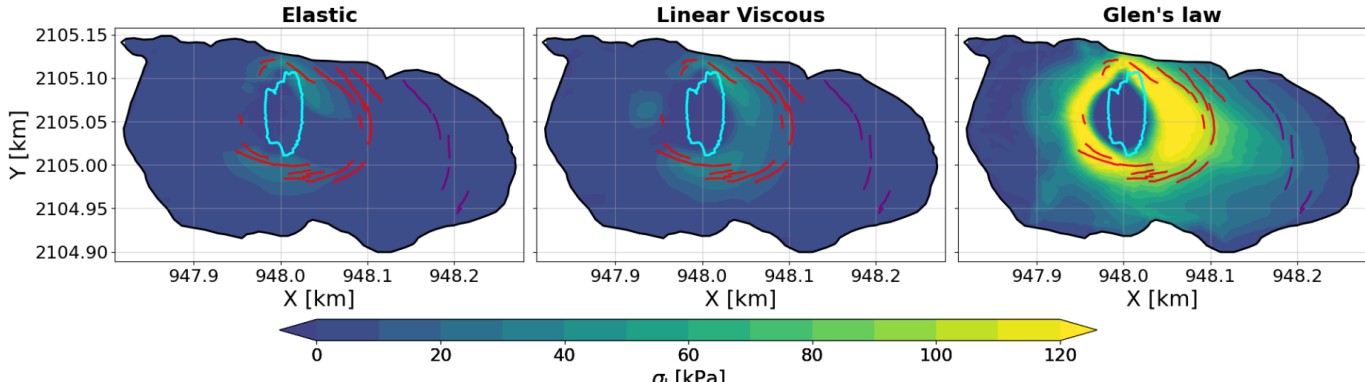

**Figure 3.** Maximum principal stress anomaly modeled at the surface due to the presence of an empty cavity and for the three constitutive laws considered: linear elastic (left), linear viscous (middle), and non-linear Glen's law (right). The cavity contour is shown in cyan. Circular crevasses mapped in summer 2011 are marked in red, while crevasses likely unrelated to cavity drainage are marked in magenta.

the fact that the computed stress field is independent of the constitutive law parameters stems from the fact that our experiments are force-driven rather than displacement or velocity-driven.

Regardless of the constitutive law, the presence of an empty cavity induces an anomaly in the surface stress field, forming a crown of elevated tensile stresses around the cavity. This crown is continuous when stresses are computed using Glen's law and shows a good agreement with the mapped circular crevasses. In contrast, it appears discontinuous over narrow regions in the linear viscous case and over broader regions in the elastic case. In the latter, some circular crevasses are located in areas without any apparent stress anomaly. The magnitude of the anomalies is similar for the elastic and linear viscous cases, with peak stress anomalies reaching approximately $40\,\mathrm{kPa}$ for the linear elastic response and $50\,\mathrm{kPa}$ for the linear viscous response. In contrast, Glen's law results in significantly larger stress anomalies that extend over a much wider area. In fact, except for the very upstream and downstream regions, the entire glacier surface is affected by the presence of an empty cavity. In this case, the highest recorded stress anomaly reaches approximately $150\,\mathrm{kPa}$.

### 4.2 Modeled surface displacements

Figure 4 compares the three components of total displacement measured at stakes between 14 September and 06 October 2010 (i.e., during the 2010 pumping sequence) with those simulated using the elastic framework with $E = 1\,\mathrm{GPa}$ and Glen's law with the spatially-distributed $T$ field. Table 3 summarizes the minimum and maximum displacements measured and simulated above the cavity over the three periods. While both constitutive laws yield similar displacement patterns indicative of subsidence of the cavity roof, the amplitude of the elastic displacements is systematically two orders of magnitude too low: a few millimeters whereas measurements indicate displacements of the order of $10\,\mathrm{cm}$ over the period. As reported above, simulations with $E = 9\,\mathrm{GPa}$ yield displacements that are nine times smaller and therefore remain systematically below $1\,\mathrm{mm}$. In contrast, in line with Gagliardini et al. (2011), Glen's law produces displacements that are very close to that observed in the x







**Figure 4.** Total displacements in the x (top row), y (middle row) and z (bottom row) directions, obtained from (left) diagnostic simulations based on the linear elastic framework with $E = 1$ GPa, (middle) transient simulation using Glen's law with the spatially distributed $T$ field, and (right) linearly interpolated from total station measurements at stakes between 14 September 2010 and 06 October 2010. Note that displacement units are in millimeters for the elastic case and in centimeters for the other cases. In the right column, black dots indicate positions of stakes. The cavity contour is shown in black.

and y directions, although vertical displacements are slightly overestimated by the model. The peak of vertical displacement in the model is also slightly downstream compared to that measured at the stakes, which may indicate inaccuracies in the reconstructed cavity shape. Given the temperature field in the vicinity of the cavity, replacing the spatially-distributed $T$ field by a uniform temperature of $T = 0$ C (resp. $T = -2$ C) produces slightly more (resp. slightly less) displacements (Table 3).

Two similar figures, covering the periods from 09 to 28 September 2011 (when the cavity was filled with water and under 330 pressure) and from 28 September to 21 October 2011 (spanning the 2011 pumping sequence), are available in the Supplement





(Figs S5 and S6). Overall, the same observations hold as for the 2010 pumping sequence described above. We note, however, that the overestimation of the rate of subsidence of the cavity simulated by the viscous model is higher for the 2011 pumping sequence than for the 2010 pumping sequence. This could be attributed to uncertainties regarding the cavity shape that increase over time, particularly with the detachment of ice blocks from the roof. This detachment, observed in the field but not

accounted for in the model, would have lightened the roof of the cavity, thereby reducing the gravitational forces compared to those estimated in the model. We also note that the model based on Glen's law is less effective at capturing the cavity roof uplift observed in September 2011 (although the measured displacement field is significantly noisier, with few stakes showing subsidence while neighboring stakes exhibit uplift) than it is at representing the subsidence during the pumping sequences. The elastic model, on the other hand, does capture the uplift but, as before, underestimates the displacement by one to two orders

of magnitude.

  To summarize, these results show that the deformation observed at the glacier surface in response to pressure variations in the cavity is of viscous type, with the elastic component being largely negligible over the time scales of interest. As a consequence, the results presented in the following of this section rely solely on Glen's law.

### 4.3    Failure criterion and Stress Threshold

In Section 4.1, we analyzed the diagnostic surface stress field under the assumption of instantaneous cavity drainage. This represents an unrealistic worst-case scenario because: (1) in reality, cavity drainage is not instantaneous (for example, the 2010 pumping operations lasted 50 days); and (2) the resulting stress field is transient, and stresses are redistributed over time by viscous deformation. Here, we present results from transient simulations performed with Glen's law, which capture the time-dependent evolution of stresses in response to water pressure fluctuations in the cavity. Figure 5 shows the equivalent stress at

the surface computed at the end of the 2010 drainage with the four failure criteria introduced in Sect. 3.2.

  Both the MPS and Hayurst criteria are characterized by a crown of elevated $\sigma_{\mathrm{eq}}$ surrounding the cavity roof, which matches relatively well the circular crevasses mapped at the surface. The similarity between the patterns obtained with the MPS and Hayurst criteria arises from the fact that the latter is a linear combination of several criteria, with the highest weight $\alpha$ assigned to the MPS. This also explains why, under the Hayhurst criterion, the region above the cavity roof exhibits reduced but nonzero

$\sigma_{\mathrm{eq}}$, whereas the MPS criterion produces a circular region of strictly zero $\sigma_{\mathrm{eq}}$ in this area. Indeed, the vM criterion, which is also included in the Hayhurst equivalent stress calculation with a weighting coefficient $\beta$, produces very high $\sigma_{\mathrm{eq}}$ in this region. These areas where the vM criterion produces the highest $\sigma_{\mathrm{eq}}$ are mostly free of crevasses. Similarly, although the Coulomb criterion results in overall lower $\sigma_{\mathrm{eq}}$ values compared to the vM criterion, the same observation holds: crevasses are found in regions of relatively low $\sigma_{\mathrm{eq}}$, while the highest $\sigma_{\mathrm{eq}}$ are found over the crevasse-free cavity roof. This stands in strong contrast

to the Hayhurst and MPS criteria, for which crevasses are systematically found in regions of highest $\sigma_{\mathrm{eq}}$ .

  Figure 6 illustrates the evolution of surface equivalent stress along three transects throughout the 2010 pumping sequence for the four considered failure criteria. A corresponding figure for the period between the 2010 and 2011 pumping operations, during which the cavity refilled, is available in the Supplement (Fig. S7). Note that although the MPS criterion considers only the positive values of $\sigma_{\mathrm{I}}$ according to Eq. (3), negative values are kept here to highlight areas of compressive stresses (first row




**Table 3.** Minimum and maximum displacements above the cavity in the x (top lines), y (middle lines), and z (bottom lines) directions obtained over three periods from diagnostic simulations based on the linear elastic framework, transient simulation using Glen's law, and total station measurements at stakes. All displacements are given in centimeters.

| Period | | 14 Sept. to 06 Oct. 2010 | | 09 Sept. to 28 Sept. 2011 | | 28 Sept. to 21 Oct. 2011 | |
|---|---|---|---|---|---|---|---|
| Law | Parameter | Min disp. | Max. disp. | Min disp. | Max. disp. | Min disp. | Max. disp. |
| Elastic | $E = 1$ GPa | 0.0 | 0.2 | −0.1 | 0.0 | −0.1 | 0.3 |
| | | −0.1 | 0.1 | 0.0 | 0.0 | −0.1 | 0.2 |
| | | −0.3 | 0.0 | 0.0 | 0.1 | −0.6 | 0.0 |
| | $E = 9$ GPa | 0.0 | 0.0 | 0.0 | 0.0 | 0.0 | 0.0 |
| | | 0.0 | 0.0 | 0.0 | 0.0 | 0.0 | 0.0 |
| | | 0.0 | 0.0 | 0.0 | 0.0 | −0.1 | 0.0 |
| Glen's | $T = f(x,z)$ | −13.0 | 2.5 | −3.8 | −2.1 | −10.9 | 0.7 |
| | | −4.4 | 8.7 | 0.1 | 0.9 | −3.5 | 6.9 |
| | | −27.0 | −1.9 | −1.3 | 0.0 | −19.7 | −2.4 |
| | $T = 0$ C | −15.5 | 3.3 | −4.3 | −2.2 | −12.6 | 0.9 |
| | | −5.5 | 10.7 | 0.1 | 0.9 | −4.2 | 8.1 |
| | | −33.0 | −2.2 | −1.4 | 0.1 | −22.9 | −2.8 |
| | $T = -2$ C | −11.6 | 1.6 | −3.6 | −2.1 | −9.9 | 0.3 |
| | | −3.9 | 7.7 | 0.1 | 0.8 | −3.2 | 6.3 |
| | | −23.2 | −1.9 | −1.2 | 0.0 | −17.6 | −2.4 |
| Observations | - | −12.0 | 1.0 | −4.8 | −0.8 | −16.0 | −0.5 |
| | | −3.5 | 6.8 | −0.8 | 2.2 | −6.2 | 3.3 |
| | | −18.4 | −2.1 | −6.8 | 6.1 | −6.7 | 0.8 |

of Fig. 6). When discovered the cavity was under pressure as demonstrated by the strong positive maximum principal stress ($\sim 100$ kPa) prevailing over the cavity roof at the very beginning of the simulation. Immediately after the drainage begins, the stresses over the cavity roof shift to slightly compressive, while the crown of tensile stresses forms around the roof. These tensile stresses continue to increase throughout the pumping operation, reaching their highest level at the end.

Consistent with what has been inferred from Fig. 5, $\sigma_{\mathrm{eq,H}} \approx \sigma_{\mathrm{I}}$ in regions where $\sigma_{\mathrm{I}}$ is high enough (i.e., $\sigma_{\mathrm{I}} \sim 100$ kPa
or greater). This is because, in these regions, the value of the von Mises equivalent stress itself is primarily ruled by $\sigma_{\mathrm{I}}$. In contrast, in regions where $\sigma_{\mathrm{I}}$ is lower, the values of $\sigma_{\mathrm{II}}$ and $\sigma_{\mathrm{III}}$ become important in the computation of $\sigma_{\mathrm{eq,vM}}$. As a result, in these areas $\sigma_{\mathrm{eq,H}}$ deviates from $\sigma_{\mathrm{I}}$. It is also interesting to note that the equivalent stress profiles obtained at the surface with the Coulomb and von Mises criteria are somewhat similar in shape but different in magnitude, with the von Mises critetion producing much higher equivalent stress.



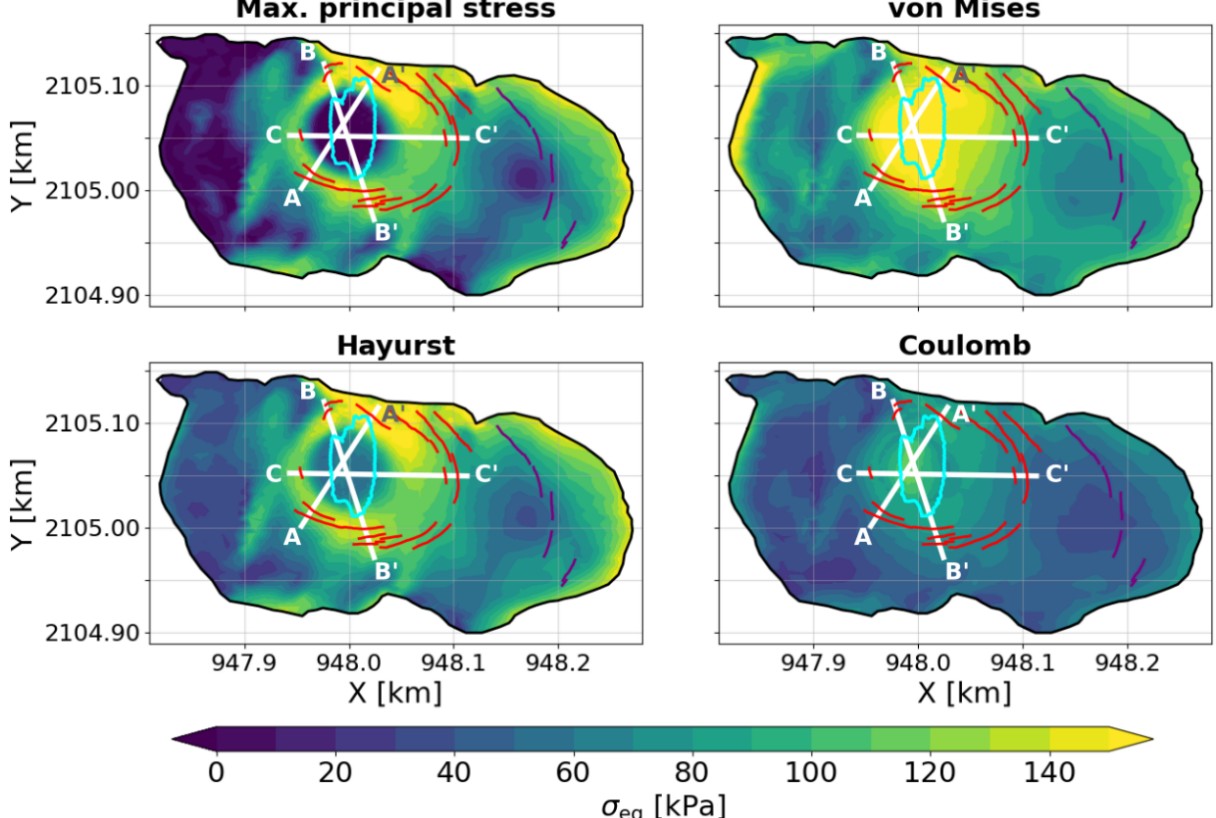

**Figure 5.** Equivalent stress at the surface at the end of the 2010 pumping operation, computed using the four failure criteria presented in Sect. 3.2. The cavity contour is shown in cyan. Circular crevasses mapped in summer 2011 are marked in red, while crevasses likely unrelated to cavity drainage are marked in magenta. Transects AA', BB', and CC', along which equivalent stresses are plotted in Fig. 6, are indicated in white.

Crevasses are systematically found in areas corresponding to peaks of $\sigma_I$ (i.e., the crown of high tensile stresses described above). Only the upstream end of the CC' transect and, to a lesser extent, the downstream end of BB' transect exhibit a slight shift between crevasse positions and stress peaks. As for vertical displacement peaks, this may reflect uncertainties in the exact shape of the cavity. In addition, the value of equivalent stress at which crevasses tend to occur is remarkably robust: crevasses are found in places where $\sigma_I$ is between 100 and 130 kPa. Only the crevasse at the upstream end of the AA' profile seem to

occur at higher stress ($\sigma_I \approx 165$ kPa) if we focus on the stress field computed at the end of the pumping operation. However, this crevasse could have started to initiate earlier in the pumping operation, around September 15, when peak stresses were lower, although this observation was not reported by those operating in the field. In contrast, as inferred above from Fig. 5, no clear correlation is observed between the presence of crevasses and singularities in equivalent stress when computed using the von Mises or Coulomb criterion.





## 4.4 Damage feedback on viscosity


The results presented above suggest that a maximum principal stress criterion, combined with a stress threshold $\sigma_{\text{th}}$ of $100 - 130$ kPa, is the most suitable approach to represent the initiation of crevasses in numerical models. Although this provides crucial qualitative information on whether crevasses are likely to form in a given region, it does not offer quantitative insights, such as crevasse concentration to be expected locally. In the framework of CDM, this aspect is quantified through the damage

variable $D$, which ranges between $0$ (undamaged) and $1$ (fully damaged), and directly influences ice viscosity through Eq. (A9). The rate at which damage grows is proportional to the gap between the local value of the equivalent stress and the stress threshold (Eqs. (A10) and (A11)). The proportionality constant, known as damage enhancement factor $B$, is an unknown parameter that directly affects the degree of ice softening expected locally. The prognostic simulations with damage introduced in Sect. 3.3 aim to determine whether $B$ can be constrained using displacement data from the cavity roof monitored between

August 2010 and September 2013. Figure 7 compares the mean vertical velocity measured at stakes above the cavity roof with simulations using Glen's law, both without and with damage feedback on viscosity. Although the von Mises and Coulomb criteria should be disregarded on the basis of the results presented above, simulations including damage consider all of the four failure criteria. Each panel corresponds to one of the eight combinations of parameters $(B, \sigma_{\text{th}})$ tested (Table 2). We emphasize that, while blue dots represent mean vertical velocities derived from measurements performed at stakes above the cavity, error

bars do not represent measurement precision but the range between the fastest and slowest mean vertical velocities recorded across these stakes during each measurement period (Fig. 7). Consistent with the results reported in Sect. 4.2 and regardless of whether damage is accounted for, all prognostic simulations capture remarkably well both the acceleration of cavity roof subsidence during the 2010 pumping and the subsequent slowdown associated with the cavity refill in late spring/summer 2011. As already mentioned above, the slow uplift of the cavity roof observed in late summer 2011 is less well captured, while the

rate of subsidence associated with the 2011 pumping operation is exaggerated in the model, likely due to errors in the cavity roof thickness following the detachment of ice blocks. The second striking result is that, independently of the parameter pair $(B, \sigma_{\text{th}})$, the damage model based on a MPS criterion, and to a lesser extend the one based on the Hayurst criterion, produces the same behavior of the cavity roof as the model without damage. This occurs because, with these two criteria, damage develops in the crown surrounding the cavity roof, but it does not have time to advect to the roof itself due to the very low

surface velocity. As a result, the viscosity of the cavity roof remains unaffected, making it impossible to constrain the value assigned to $B$ based on this experiment.

Although both criteria should be discarded based on previous results, it is still noteworthy that, because the von Mises and Coulomb criteria produce damage directly on the cavity roof, the deformation of the latter is strongly affected by the inclusion of damage when using either of these criteria, provided that the stress threshold is not too high and the parameter $B$

is sufficiently large. More specifically, since $\sigma_{\text{eq,C}}$ never exceeds $150$ kPa, all combinations with $\sigma_{\text{th}}$ higher than this value (combinations 4, 6 and 8) result in no damage under the Coulomb criterion. In contrast, peaks of $\sigma_{\text{eq,vM}}$ exceeding $200$ kPa, all simulations relying on the von Mises criterion produce damage on the cavity roof, which then deforms more readily, particularly when the gap between $\sigma_{\text{eq}}$ and $\sigma_{\text{th}}$ (e.g. combinations 2, 3, 7) and/or $B$ is large (e.g. combinations 1, 5, 7, 8).



## 5 Discussion

### 5.1 Elastic or Viscous ?


Due to the nature of the boundary conditions (i.e., the experiment is fully force-driven, with no imposed displacements), the instantaneous stress field obtained in response to a given perturbation of water pressure in the cavity is mainly dictated by the force balance and is independent of Young's modulus in the elastic framework and of fluidity in the viscous framework (Figs S2 to S4). However, the resulting strain and strain rates are highly sensitive to these parameters and can thus be used as

control variables for comparison with observations. This stands in sharp contrast to studies that infer stress states in crevassed areas from observed surface strain or strain rates, where the inferred stresses are highly sensitive to assumptions about ice stiffness or fluidity (Vaughan, 1993; Ultee et al., 2020; Grinsted et al., 2024; Wells-Moran et al., 2025). Of course, in transient viscous simulations, feedback mechanisms occur: differences in velocity responses for different fluidities lead to differences in geometry evolution, which in turn affect subsequent stress fields, and so on.

Although the instantaneous stress response within a given constitutive law is independent of the fluidity or stiffness parameter, different constitutive laws yield different stress responses. This difference arises primarily from the assumption of incompressibility in the viscous framework versus compressibility in the elastic framework, as well as from the non-linearity of Glen's law. This raises the question of the nature of the mechanical response that should be considered when modeling the initiation of a crevasse in virgin ice. Using observations of the 2015 eastern Skaftá cauldron collapse (Vatnajökull ice cap,

Iceland) to infer ice tensile strength from observed crevasses, Ultee et al. (2020) estimate a tensile strength of the order of $1\,\mathrm{MPa}$, an order of magnitude higher than our findings. Their estimate relies on the assumption that the surface deformation in response to subglacial lake drainage is predominantly elastic, at least during the first three days. This assumption is based on the argument that the system operates near the Maxwell time, for which they adopt a representative value of $11\,\mathrm{h}$, where the Maxwell time is given by

$$\tau_{\mathrm{m}} = \frac{2\eta(1+\nu)}{E}. \tag{6}$$

However, as pointed out in earlier work (e.g., Podolskiy et al., 2019; Hageman et al., 2024), the non-linear nature of Glen's law results in highly heterogeneous Maxwell times, which can be particularly short in regions of high deviatoric stresses and/or high temperature. For example, Fig. 8 shows the distribution of the Maxwell times at the surface and along a vertical profile of the Tête Rousse glacier, assuming an empty cavity, Glen's flow law, and a Young's modulus of $E = 9\,\mathrm{GPa}$. Maxwell times span

four orders of magnitude: they are on the order of an hour around the cavity, where deviatoric stresses are high, while they reach several months in low-stress regions. Assuming $E = 1\,\mathrm{GPa}$ would increase Maxwell times by an order of magnitude (Fig. S8). Given the much larger spatial scale of the Skaftá cauldron compared to our study case (cauldron is $2.7\,\mathrm{km}$ in diameter with an estimated roof thickness of $\sim 300\,\mathrm{m}$) and a recorded peak subsidence rate of $3\,\mathrm{m\,h^{-1}}$ (Ultee et al., 2020), it is likely that Maxwell times near the cauldron were even shorter than an hour in their case. In other words, we argue that Ultee et al. (2020)

significantly underestimated the viscous contribution to the observed surface deformation, leading to an overestimation of stress and, as a consequence, an overestimation of ice tensile strength. In contrast, our results suggest that crevasse initiation in




initially intact ice is governed by viscous processes. Another key finding of the present study is the necessity of accounting for the non-linearity of Glen's law, as it leads to higher and more localized stresses compared to a linear viscous law. Indeed, since effective viscosity decreases in regions experiencing high deviatoric stresses, these areas become less capable of sustaining
stress. As a result, the stress is redistributed to surrounding regions where the ice is stiffer.

## 5.2  What failure criterion ?

In our study, the MPS criterion proves to be the most effective predictor of crevasse initiation. The Hayhurst criterion also performs well, primarily because, in crevassed areas, the Hayhurst equivalent stress nearly reduces to the maximum principal stress. In contrast, the Coulomb and von Mises criteria do not correlate well with mapped surface crevasses. While our result is
consistent with the widely accepted notion that Mode I fracture (opening) is the dominant mechanism for crevasse formation in most settings (Colgan et al., 2016), it challenges findings based on glacier-scale in situ and remote sensing observations (Vaughan, 1993; Grinsted et al., 2024; Wells-Moran et al., 2025). These studies apply a similar approach, converting observed surface strain rates into deviatoric stresses using Glen's law, and constructing failure envelopes that enclose stress states associated to uncrevassed areas while excluding stress states associated to crevassed areas. Applying this method across 17 polar
and alpine locations, Vaughan (1993) found that the Coulomb and von Mises criteria, combined with a failure strength between 90 to 320 kPa, best matched observations. Comparable results were reported by Wells-Moran et al. (2025) for Antarctic Ice Shelves, who refined the failure strength range to 202-263 kPa (for $n = 3$). Similarly, Grinsted et al. (2024) applied the method to the Greenland Ice Sheet and found that a von Mises criterion with a failure strength of $265 \pm 73$ kPa provides an adequate fit to the data. As discussed by Wells-Moran et al. (2025), we suggest that these findings are influenced by the fact that they
rely on observations from settings where isotropic pressure is negligible compared to deviatoric stresses. As a result, the role of isotropic pressure in preventing crevasse opening cannot be used to rule out pressure-independent criteria. For example, as Wells-Moran et al. (2025) point out for Antarctic ice shelves, a von Mises criterion would likely not fit basal crevasse data well if such data were available, since overburden pressure is expected to act against crevasse opening. In our case study, although we focus on surface stresses, the subsidence of the cavity roof induces a stress pattern in which the surface isotropic pressure
deviates significantly from zero (Fig. 9). It is indeed important to keep in mind that isotropic pressure, defined as the mean of the Cauchy normal stresses, generally differs from the overburden pressure $\rho_i g(z_s - z)$. Specifically, the surface pressure over the cavity roof shifts from slightly tensile (i.e., negative) at the beginning of the simulation (Fig. 9b), when the cavity is pressurized, to largely compressive (i.e., positive) by the end of the 2010 pumping operation, when the cavity is empty (Fig. 9a,b). In this region, no crevasses are observed, yet the von Mises criterion still predicts high equivalent stresses due to its
pressure independence. A similar behavior is observed for the Coulomb criterion, which is mostly (but not fully) pressure-independent given the low value of $\mu = 0.1$ adopted here. This finding contradicts the applicability of pressure-independent criteria and instead supports the concept of a failure strength that increases with pressure, consistent with the experimental results of Nadreau and Michel (1986) for confining pressure values typically found in glaciers.

Interestingly, Grinsted et al. (2024) found that a Schmidt-Ishlinsky criterion outperforms the von Mises criterion in de-
scribing glacier ice failure. The Schmidt-Ishlinsky criterion is essentially a MPS criterion, except that it considers deviatoric



stresses rather than Cauchy stresses. However, since isotropic pressure is arguably small in their case, the MPS and Schmidt-Ishlinsky criteria would be expected to yield similar results. The stress threshold associated with this latter criterion in their study, $158 \pm 44$ kPa, is consistent with our estimate of $100$–$130$ kPa.

### 5.3 From crevasse initiation at the surface to full-thickness fracturation

As in most studies aiming to assess a failure criterion and stress threshold for crevasse initiation (Vaughan, 1993; Ultee et al., 2020; Grinsted et al., 2024; Wells-Moran et al., 2025), we focused on the stress fields at the surface, where crevasses are commonly conceptualized to initiate, at least in ablation areas (Colgan et al., 2016). We emphasize that our study focuses specifically on crevasse initiation and not on fracture propagation, which involves different mechanisms. In particular, our finding that crevasse initiation in intact ice is governed by viscous rather than elastic processes does not extend to fracture

propagation. Indeed, once a crevasse reaches a critical depth, stress concentrations at the crack tip become the dominant control of the propagation, and elastic processes take over as the primary mechanism, although viscous deformation may still contribute (Hageman et al., 2024). Because fractures are not explicitly represented in the computational domain, models based on a continuum approach cannot directly capture these stress concentrations. Even if fractures were explicitly resolved, the stress distribution near crevasse tips would be highly sensitive to mesh resolution. A common alternative to indirectly account

for these stress singularities while keeping a continuum framework is to apply Linear Elastic Fracture Mechanics (LEFM), which requires the presence of small initial cracks as starting conditions (Krug et al., 2014). Studies that have attempted to model calving directly within a viscous framework, without this LEFM layer, have found that a von Mises criterion produces the most realistic behavior in terms of calving front evolution (Choi et al., 2018; Mercenier et al., 2019). We believe that this is because only a pressure-independent criterion can allow a water-free fracture to propagate through the full thickness of the ice

body when stress concentrations are not explicitly accounted for. Although this may be an acceptable approach to implement an 'averaged' calving rate in large-scale ice sheet models, we argue that it overlooks the fundamental physics of ice fracture and is therefore not universally applicable for forecasting specific ice mass collapse events. Instead, our findings support modeling approaches that couple a viscous flow model to describe crack nucleation with an LEFM approach to simulate subsequent fracture propagation (e.g., Krug et al., 2014; Yu et al., 2017; Zarrinderakht et al., 2022). Although still in its early stages and

so far limited to simplified synthetic cases, the phase-field method appears to be another promising approach, offering the potential to model crack initiation, propagation, and branching within a unified framework (e.g., Sun et al., 2021; Clayton et al., 2022; Sondershaus et al., 2023).

## 6 Conclusions

In this study, we use the rich dataset from the careful monitoring of the artificial drainage of a water-filled cavity on Tête
Rousse Glacier to investigate crevasse initiation processes. Specifically, we use numerical experiments to determine the failure criterion and its associated stress threshold that best match the observed circular crevasses that appeared following the 2010





pumping operation. In contrast to previous studies that inferred stresses from observed deformations, our good knowledge of the cavity geometry and of the evolution of water level enables a direct computation of stresses from the force balance.

It turns out that the stress field assuming a non-linear viscous rheology best matches the observed data as it induces a
concentration of the stresses on the edges of the cavity where ice is stiffer, an effect that is not observed with a linear viscous law. Furthermore, even in the extreme and unrealistic scenario of an instantaneous drainage of the cavity, we show that the elastic component of the total deformation is largely negligible compared to the viscous component. In fact, due to the non-linearity of the flow law, the Maxwell time is highly heterogeneous, spanning several orders of magnitude. In regions with large deviatoric stresses, the Maxwell time can be on the order of an hour or less, causing the viscous response to rapidly dominate
over the elastic response.

Adopting the usual non-linear Glen's law with $n = 3$, we show that a Maximum Principal Stress criterion, combined to a stress threshold of approximately $100$ to $130$ kPa, works well for modeling crevasse initiation. The specific stress pattern associated with such a subsidence event combined to the absence of crevasses on the cavity roof allows us to rule out pressure-independent criteria, such as the von Mises criterion, which observation-based studies applied to settings with small isotropic
pressure cannot do. Regarding fracture propagation through the full ice thickness, our results support modeling approaches that couple a viscous flow model for crevasse nucleation with a linear elastic fracture mechanics (LEFM) approach for their propagation. Although recent applications of the phase field method to idealized glaciological problems show promise, the method remains in an early stage and requires significant further development.

Our study focuses on the stress fields at the surface, where crevasses are expected to initiate in pure ice settings. In accu-
mulation areas, however, crevasses are expected to form within the firn layer rather than in ice. Firn is characterized by its compressibility and by a viscosity that decreases rapidly and non-linearly with decreasing density (Gagliardini and Meysson-nier, 1997). As a result, unlike in ice, peaks of tensile stress in firn are not located at the surface but rather a few meters below, where the stiffer firn can sustain greater stress than the highly deformable surface layer, and where the overburden pressure is still moderate. We anticipate that the Maximum Principal Stress criterion remains valid in firn, but that the stress threshold
must be parameterized as a function of density. This parameterization is currently underway.

*Code and data availability.* The Elmer/Ice code is publicly available on GitHub at https://github.com/ElmerCSC/elmerfem (last access: May 2025). All simulations were carried out using Elmer/Ice version 9.0 (Rev:0f18c8f). All materials required to reproduce the simulations presented in this paper are available in JB's GitHub repository: https://github.com/jbrondex/TeteRousse_TC2025.

## Appendix A: Model equations

Here, we present the mathematical model. The computational domain is three-dimensional, with the coordinate system defined as follows: the $x$-axis corresponds to the west-east direction, the $y$-axis to the south-north direction, and the $z$-axis to the vertical, pointing upwards. The variable $t$ represents time.



## A1 Field equations and constitutive laws

The distribution of stresses in a body subjected to body forces and external loads is governed by the momentum balance.
Neglecting inertial effects, the latter reads

$$\operatorname{div}\boldsymbol{\sigma} + \rho_{\mathrm{i}}\mathbf{g} = 0, \tag{A1}$$

where $\boldsymbol{\sigma}$ is the Cauchy stress tensor, $\rho_{\mathrm{i}}$ the ice density, and $\mathbf{g}$ the gravitational acceleration vector. The Cauchy stress tensor can
be decomposed into its deviatoric $\mathbf{S}$ and isotropic $p\mathbf{I}$ parts as $\boldsymbol{\sigma} = \mathbf{S} - p\mathbf{I}$, with $p$ the isotropic pressure and $\mathbf{I}$ the identity matrix.
In this work, we adopt the standard convention in fluid mechanics whereby the pressure $p$ is defined as positive in compression
and negative in tension. Equation (A1) alone is not sufficient to determine $\boldsymbol{\sigma}$, as it consists of three scalar equations for the six
independent components of the stress tensor. To close the system, a constitutive law must be prescribed. Below, we detail the
linear elastic and viscous constitutive laws that have been used in this study.

### A1.1 Linear elastic framework

Assuming deformations are small, the linear elastic behavior of ice is described by Hooke's law, which relates the Cauchy
stress tensor $\boldsymbol{\sigma}$ to the linearized strain tensor $\boldsymbol{\epsilon}$ as follows:

$$\boldsymbol{\sigma} = \lambda\operatorname{tr}(\boldsymbol{\epsilon})\mathbf{I} + 2\mu\boldsymbol{\epsilon}, \tag{A2}$$

where $\lambda$ and $\mu$ are the first and second Lamé parameters given by:

$$\lambda = \frac{E\nu}{(1+\nu)(1-2\nu)}, \quad \mu = \frac{E}{2(1+\nu)}, \tag{A3}$$

where $E$ is Young's modulus and $\nu$ Poisson's ratio. The components of the linearized strain tensor are expressed as a function
of the components of the displacement vector $\mathbf{u}$ as follows:

$$\epsilon_{ij} = \frac{1}{2}\left(\frac{\partial u_i}{\partial x_j} + \frac{\partial u_j}{\partial x_i}\right). \tag{A4}$$

The system of Eqs. (A1)–(A2)–(A4) is now closed and can be solved for the three components of the displacement field. The
evolution of the geometry follows from the computed displacement field.

### A1.2 Viscous framework

The viscous behavior of ice is described by Glen's flow law (Glen, 1955), which expresses the deviatoric stress tensor $\mathbf{S}$ as a
function of the strain rate tensor $\dot{\boldsymbol{\epsilon}}$:

$$\mathbf{S} = 2\eta\dot{\boldsymbol{\epsilon}}, \tag{A5}$$

where the strain rate tensor components are defined as a function of the components of the velocity vector $\mathbf{v}$ as:

$$\dot{\epsilon}_{ij} = \frac{1}{2}\left(\frac{\partial v_i}{\partial x_j} + \frac{\partial v_j}{\partial x_i}\right). \tag{A6}$$





The effective viscosity $\eta$ is expressed as:

$$\eta = \frac{1}{2}(E_{\mathrm{D}}A)^{-1/n}\dot{\epsilon}_{\mathrm{e}}^{(1-n)/n}, \tag{A7}$$

where $\dot{\epsilon}_{\mathrm{e}} = \sqrt{\frac{1}{2}\dot{\varepsilon}_{ij}\dot{\varepsilon}_{ij}}$ is the second invariant of the strain rate, $n$ the flow law exponent, $E_{\mathrm{D}}$ an enhancement factor, and $A$ the fluidity parameter. The temperature dependence of ice fluidity follows an Arrhenius-type relationship:

$$A = A_0 \exp\left(-\frac{Q}{RT}\right), \tag{A8}$$

with $A_0$ a reference fluidity, Q the activation energy, R the gas constant, and T the temperature (in K). In addition, for simulations that are run with the damage module activated, the feedback of damage on viscosity is accounted for via the enhancement factor through the following relationship:

$$E_{\mathrm{D}} = \frac{1}{(1-D)^n}, \tag{A9}$$

where $D$ is the damage variable. Following Krug et al. (2014), damage evolution is governed by the advection equation:

$$\frac{\partial D}{\partial t} + \mathbf{v}\nabla D = \begin{cases} B\chi & \text{if } \chi > 0, \\ 0 & \text{otherwise .} \end{cases} \tag{A10}$$

Here, $B$ is a damage enhancement factor, and $\chi$ is the damage criterion, defined as:

$$\chi = \frac{\sigma_{\mathrm{eq}}}{1-D} - \sigma_{\mathrm{th}}, \tag{A11}$$

where $\sigma_{\mathrm{th}}$ is the stress threshold and $\sigma_{\mathrm{eq}}$ the equivalent stress. As thoroughly explained in Sect. 3.2, the expression of $\sigma_{\mathrm{eq}}$ as a function of the stress tensor invariants depends on the selected failure/damage criterion, to which the stress threshold must also be adapted. The formulation in Eq. (A11) implies that ice already experiencing damage is more susceptible to further damage compared to intact ice. This can be interpreted as an empirical approach to account for stress concentration at crevasse tips, which are not explicitly modeled. However, we stress that this implementation alone is insufficient to propagate damage at depth when a maximum principal stress criterion is adopted. A healing criterion could easily be incorporated into this framework (e.g., Pralong and Funk, 2005), but it is not considered here for the following reasons: (i) very little information is available in the literature on the dynamics of healing in ice (Colgan et al., 2016), and (ii) given the very low flow velocities and the short duration of our simulations, we assume that damaged ice largely remains in regions where damage was initially created.

The system of Eqs. (A1)–(A5)–(A6) is not closed. This is because, unlike Hooke's law, Glen's flow law accounts only for the deviatoric part of the Cauchy stress tensor, leaving the isotropic pressure $p$ as an additional unknown. To fully close the system, we incorporate the mass conservation equation. In the viscous framework, ice is assumed to be incompressible, which simplifies the mass conservation equation to $\mathrm{div}\,\mathbf{v} = 0$. With this additional constraint, the system is fully determined and can be solved for the three components of the velocity field and the isotropic pressure. The velocity field governs mass transfer,





driving the delayed evolution of the geometry toward a new steady state in response to changes in external forcings. Concretely, the computational domain is vertically limited by two free surfaces, namely the upper ice/atmosphere interface $z = z_s(x, y, t)$, and the lower interface $z = z_b(x, y, t)$ that can either be grounded on the bedrock or forms walls of the cavity. An advection equation governs the evolution of these free surfaces over time, which has the general form:

$$\frac{\partial z_j}{\partial t} + v_x \frac{\partial z_j}{\partial x} + v_y \frac{\partial z_j}{\partial y} - v_z = a_j, \tag{A12}$$

where $v_x$, $v_y$, $v_z$ are the velocity components at the considered interface, $z_j(x, y, t)$ is the elevation of the considered free surface, and $a_j(x, y, t)$ is the accumulation/ablation rate. Here, the accumulation $a_s$ over the upper interface is prescribed based on field measurements, while melting/accretion $a_b$ in the cavity is neglected. For the lower interface, Eq. (A12) is completed by the non-penetration condition:

$$z_{\text{bed}}(x, y, t) \leq z_b(x, y, t) < z_s(x, y, t), \tag{A13}$$

where $z_{\text{bed}}(x, y, t)$ is the elevation of the bedrock. In the vicinity of the cavity, ice cannot penetrate the bedrock but can move away when water pressure becomes sufficiently high. Mathematically, this corresponds to solving a contact problem. At a given point on the lower interface, $\mathbf{x} = (x, y, z_b)$, ice is considered grounded if it is in contact with the bedrock and the stress exerted by the ice exceeds the water pressure, which is formulated as:

$$z_b(x, y, t) = z_{\text{bed}}(x, y, t) \qquad \text{and} \qquad \sigma_{nn}|_{z_b} > p_w(z_b, t), \tag{A14}$$

where the normal stress at the lower interface is given by $\sigma_{nn}|_{z_b} = \mathbf{n} \cdot (\boldsymbol{\sigma} \cdot \mathbf{n})|_{z_b}$, with $\mathbf{n}$ denoting the unit normal vector to the interface pointing outward, and $p_w$ representing the water pressure. Conversely, the ice is considered floating if it is either above the bedrock or in contact with it but subjected to a water pressure greater than or equal to $\sigma_{nn}|_{z_b}$, which is formulated as:

$$z_b(x, y, t) > z_{\text{bed}}(x, y, t) \qquad \text{or} \qquad \sigma_{nn}|_{z_b} \leq p_w(z_b, t). \tag{A15}$$

In Eqs. (A14) and (A15), water pressure is directly deduced from the measured water level $z_w$ (given in meters above sea level) as follows:

$$p_w(z_b, t) = \rho_w g \left( z_w(t) - z_b(x, y, t) \right), \tag{A16}$$

where $\rho_w$ is the water density.

## A2 Boundary conditions

The computational domain is delimited by three boundaries: the upper interface, the lateral boundary, and the lower interface.

### A2.1 Upper interface

Neglecting the atmospheric pressure acting on the surface, the upper boundary satisfies a stress-free condition:

$$(\boldsymbol{\sigma} \cdot \mathbf{n})|_{z_s} = 0. \tag{A17}$$



## A2.2   Lateral boundary

On the lateral boundary a no-flux/no-displacement Dirichlet condition applies in the normal direction. This condition is expressed as:

$$\mathbf{u} \cdot \mathbf{n} = 0, \tag{A18}$$

for the elastic framework, and

$$\mathbf{v} \cdot \mathbf{n} = 0, \tag{A19}$$

for the viscous framework. In the tangential direction, a free-slip/free-displacement condition is prescribed. This Neumann condition is expressed as:

$$\boldsymbol{\tau}_{\mathrm{l}} = 0, \tag{A20}$$

where $\boldsymbol{\tau}_{\mathrm{l}} = (\boldsymbol{\sigma} \cdot \mathbf{n})|_{l} - \sigma_{\mathrm{nn}}|_{l}\mathbf{n}$ is the shear stress on the lateral boundary. Note that, because the lateral boundary is very shallow (a few meters at most), the influence of this boundary condition on the simulated displacement and velocity fields is limited to the immediate vicinity of the boundary.

## A2.3   Lower interface

Two types of boundary conditions apply at the lower interface, depending on whether the ice is in contact with the bedrock or forms part of the cavity walls. Very little to no sliding is expected at the ice/bedrock interface (Vincent et al., 2015). Consequently, for the elastic framework the no-displacement Dirichlet condition $\mathbf{u}(x, y, z_b) = 0$ is imposed wherever ice is grounded. For the viscous framework, the numerical implementation of the contact problem in Elmer/Ice requires prescribing a friction law (Durand et al., 2009). We use a linear Weertman law, which relates the basal shear stress $\boldsymbol{\tau}_{\mathrm{b}} = (\boldsymbol{\sigma} \cdot \mathbf{n})|_{z_{\mathrm{b}}} - \sigma_{\mathrm{nn}}|_{z_{\mathrm{b}}}\mathbf{n}$

to the sliding velocity $\mathbf{v}_{\mathrm{b}} = \mathbf{v} - (\mathbf{v} \cdot \mathbf{n})\mathbf{n}$ as:

$$\boldsymbol{\tau}_{\mathrm{b}} = C_{\mathrm{W}}\mathbf{v}_{\mathrm{b}}, \tag{A21}$$

where the friction coefficient is set to a high value, $C_{\mathrm{W}} = 0.1 \; \mathrm{MPa\,a\,m^{-1}}$. As a result, the simulated sliding velocities remain systematically much below $1 \; \mathrm{m\,a^{-1}}$. Additionally, a non-penetration Dirichlet condition is imposed in the normal direction, expressed as:

$$\mathbf{v} \cdot \mathbf{n} = 0. \tag{A22}$$

For cavity walls, regardless of the chosen framework, water pressure is applied without shear stress, resulting in the following Neumann condition:

$$(\boldsymbol{\sigma} \cdot \mathbf{n})|_{z_{\mathrm{b}}} = -p_{\mathrm{w}}(z_{\mathrm{b}}, t)\mathbf{n}. \tag{A23}$$



*Author contributions.* OG and AG obtained funding. JB, OG and AG designed the study. JB adapted the numerical set up based on previous
work by OG. ET provided the reconstructed surface mass balance data. JB ran the numerical simulations. JB interpreted the results with help
from all co-authors. JB wrote the manuscript with contributions from all co-authors.

*Competing interests.* The authors declare that they have no conflict of interest.

*Acknowledgements.* The authors would like to thank Mylène Bonnefoy-Demongeot, Stéphane Garambois, Anatoli Legchenko, Xavier Ra-
vanat, Christian Vincent, and all others who contributed to data acquisition in the field during the numerous campaigns conducted since 2007.
This work was partially funded by the French Ministère de la Transition écologique et de la Cohésion des territoires (Ministry for Ecological
Transition and Territorial Cohesion) through its PAPROG program. During the preparation of this manuscript, we used ChatGPT to help
rephrase certain sentences in formal scientific English.



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







**Figure 6.** Evolution of surface equivalent stress along transects AA' (left column), BB' (middle column), and CC' (right column) throughout the 2010 pumping sequence, for the four failure criteria: MPS (first row), von Mises (second row), Hayhurst (third row), and Coulomb (last row). Although the MPS criterion considers only the positive values of $\sigma_I$ according to Eq. (3), negative values are kept here to highlight areas of compressive stresses. Transects are reported in Fig. 5 (white lines). Vertical lines indicate crevasses, while the grey shaded area represents the cavity roof. For each day when a profile is plotted, the corresponding water level is marked by a colored dot in Fig. 2.





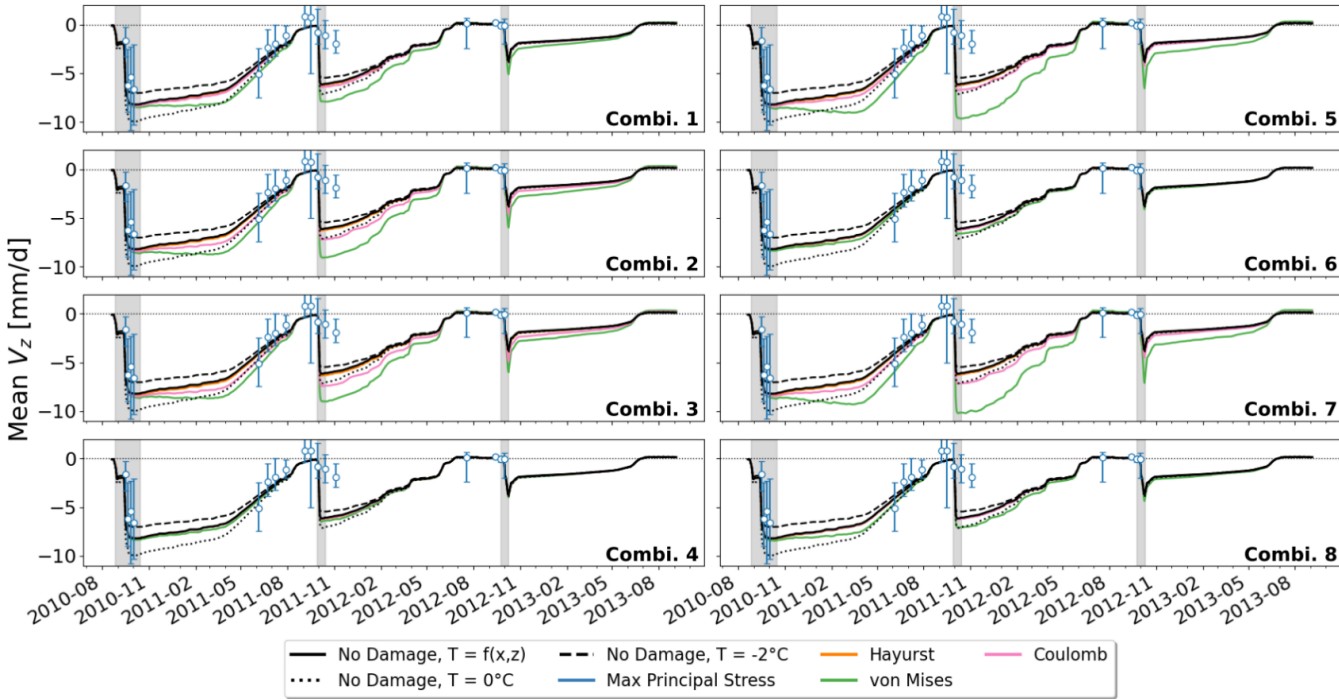

**Figure 7.** Mean vertical velocity above the cavity roof over time, simulated without damage for different temperature conditions: spatially distributed $T$ field (black solid line), uniform $T = 0$ C (black dotted line), and uniform $T = -2$ C (black dashed line). Simulations including damage are shown for the four failure criteria: MPS (blue), Hayhurst (orange), von Mises (green), and Coulomb (pink), all using the spatially distributed $T$ field. Each of the eight panels corresponds to a specific combination of parameters $(B, \sigma_{\mathrm{th}})$ as listed in Table 2. Blue dots indicate mean vertical velocities derived from stake measurements, computed between two measurement dates. Error bars do not represent measurement precision but the range between the fastest and slowest mean vertical velocities recorded across these stakes during each measurement period. Grey shaded areas correspond to pumping operations.



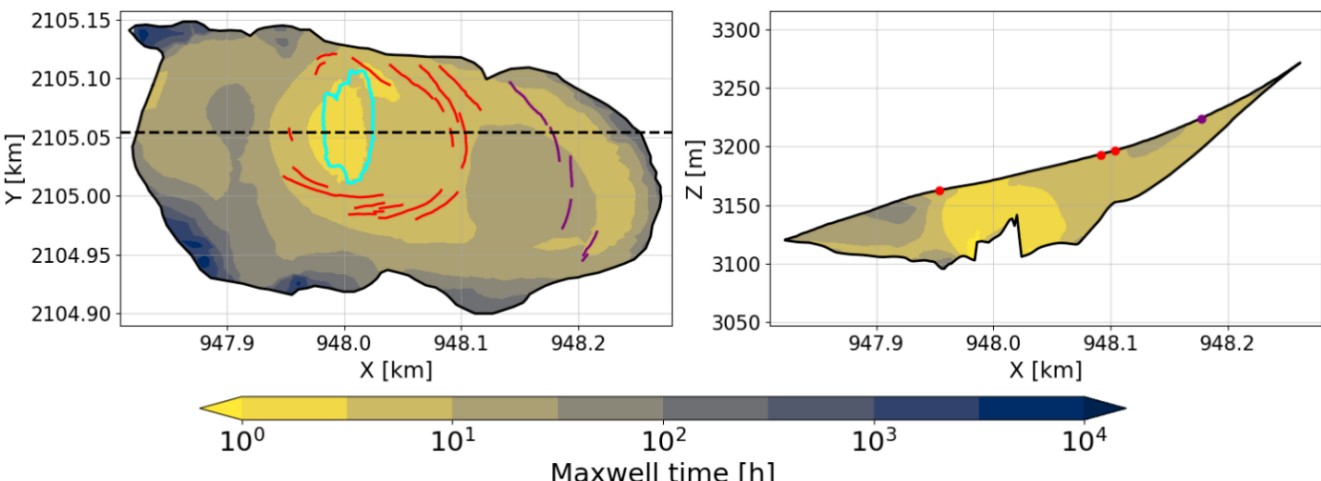

**Figure 8.** Maxwell time at the glacier surface (left) and across a vertical profile (right) evaluated for an empty cavity according to Eq. 6 with the Glen's law and for $E = 9$ GPa. The cavity contour is shown in cyan. Circular crevasses mapped in summer 2011 are marked in red, while crevasses likely unrelated to cavity drainage are marked in magenta. The transect along which the vertical profile is extracted is reported by the dashed black line in left panel.

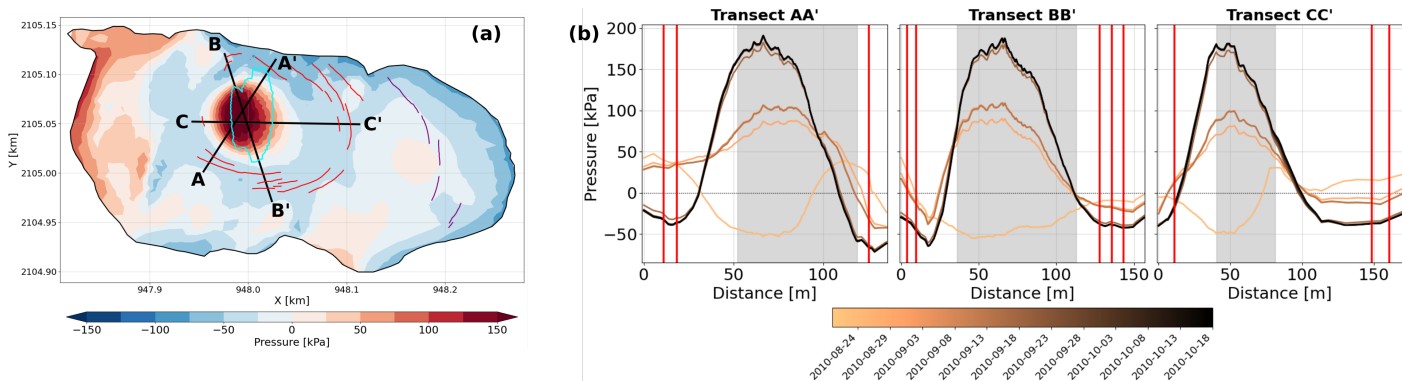

**Figure 9.** Panel (a): Same as Fig. 5, but showing pressure instead of equivalent stress. Panel (b): Same as Fig. 6 but showing pressure instead of equivalent stress. Note that we adopt the standard convention in glaciology whereby the pressure $p$ is defined as positive in compression and negative in tension.