# Peer review of "How to model crevasse initiation? Lessons from the artificial drainage of a water-filled cavity on the Tête Rousse Glacier (Mont Blanc range, France)"

_EGUsphere, 2025_

## Referee Comment (RC1)

**General Comments**

This manuscript aims to address the question: "what is the tensile strength of ice?" using a unique set of stress data collected during the drainage of a glacier cavity. The authors utilize water pressure to directly calculate the force exerted on the glacier by slowly draining the cavity, mitigating some of the uncertainties in estimated stresses introduced by assumed rheological properties. The authors then use the finite element model Elmer/Ice to replicate the drainage with different rheological properties to see which model configuration best replicates observed stresses. The authors find that a non-linear viscous flow regime defined by n=3 best matches observations.

I found parts of the paper to be a little challenging to follow due to the structure it was presented in. Some of the readability issues are in part due to the overuse of passive voice in many sections. I have made more specific comments below on suggestions for improved readability.

Overall, I think this paper is an interesting concept with a unique dataset that approaches estimating tensile strength in a novel way, but could use further refinement in the parameters selected for the model and in the organization of the paper. My primary concern is in the choice to use a linear-viscous test case when such states have not been able to be reproduced in ice in the lab, and the absence of grain size dependent viscosity when n<3. I would also like more elaboration about the applicability of these results derived from a slow-flowing temperate glacier to fast-flowing, cold Antarctic ice. With these comments addressed, I think this paper will make an excellent contribution to the literature.

**Specific Comments**

Temperate glaciers, especially those with low stresses and strains, tend to have deformation dominated by grain boundary sliding (n=1.8). For deformation regimes defined by n<3, ice viscosity is non-linearly dependent on grain size, and the flow rate parameter has units of $Pa^{-n} m^{-d} s^{-1}$ where d is the grain size exponent. In line 261, the units of the adjusted flow rate parameter for n=1 do not have a length component, suggesting the authors haven't included grain size dependence in the ice viscosity estimates. Goldsby and Kohlstedt (2001) have an equation for the flow law pre factor for n=1, but note that an n=1 flow regime has never been observed in laboratory experiments or natural terrestrial ice. Given the lack of observations of an n=1 flow regime in ice, I would recommend the authors replace the linear viscous test case with a model run with n=1.8 and a grain size dependent viscosity to make the test case more applicable to natural glacier systems. Goldsby and Kohlstedt's 2001 paper provides excellent estimates of A, n, and Q for various flow regimes, as well as modified Arrhenius relations that take into account grain size. Given the small strain rates described in the paper, I would expect grain boundary sliding to dominate over dislocation creep in the majority of the glacier, which may explain the overestimation of the subsidence rates in the n=3 regime. It also may be the case that during the drainage event, there is a shift from an n=1.8 to n=3-4 regime because of the increase in stress caused by loss of hydrostatic pressure. I wonder if there is a way to determine if this shift in n values occurs based on observed rates of subsidence over the cavity, or by crevasse opening rates? I would also be curious to see a prognostic model run with n=4, to complete the full range of non-linear viscous flow regimes for ice.

The majority of the tensile strength studies cited in this manuscript are derived from observations of cold ice, very fast moving ice, or both. I'd like to see some discussion about how tensile strength estimates might differ in temperate versus cold ice, and the applicability of the tensile strength estimates produced in this study to Greenland and Antarctica where the ice is flowing orders of magnitude faster and the deformation mechanism may be different. Currier and Schulson (1982) suggest tensile strength also has a grain size dependency, which may also affect the estimated tensile strength values of this study compared to estimates from Greenland or Antarctica.

Many sections of this paper alternate between passive (e.g. simulations were conducted, the data were extruded) and active voice (e.g. we conduct simulations, we extrude the data), making these sections challenging to follow. I would recommend the authors switch instances of passive voice into active voice. Section 2 is especially difficult to follow given the use of passive voice, as it is challenging to determine which actions were performed by the cited studies and which actions the authors took. Section 3 is also heavy on passive voice.

Lines 21-23: "In addition, crevasses are involved in two processes that control the rate at which ice sheets lose mass into the ocean: iceberg calving (Colgan et al., 2016) and accelerated flow due to the softening of shear margins (Borstad et al., 2016)." This line could use more citations for each case

Line 33-34: add Chudley et al., 2021 to citation list

Section 3:
- I would suggest a small restructuring of sections 3.2 and 3.3 to improve readability of the methods section. Rather than having the yield criteria in their own section breaking up the two sections on modeling, it may read smoother to split the last paragraph of section 3.3.3 into a new subsection solely about modeling damage, and move the descriptions of the yield criteria to the new subsection. So the flow of section 3 would be: model description —> diagnostic and prognostic —> implementing damage into model.
- Having the equations in the appendix made this section challenging to follow. I'd recommend integrating some of the key equations such as Glens law, the Arrhenius relation, and the equation used to calculate pressure from the water drainage into the main text

Section 3.2 intro: State a brief definition of equivalent stress

Line 195: State the sign convention for principal stresses (e.g. tension positive compression negative)

Equation 1: define what *Ice* $\sigma_{III}$ is

Lines 281-282: "Water pressure is applied in the cavity, i.e. at nodes identified as being above the bedrock." Wording is confusing, as all nodes in the model domain are above the bedrock. Could be reworded as "Water pressure is applied in the cavity, i.e. at nodes on the basal boundary which are not in contact with the bedrock"

Section 4.4: The authors detail how they set up the prognostic model with the damage-dependent viscosity and talk about where damage develops with each given model, but do not show these results in data or figures. Is it possible to include a figure showing the evolution of D over time with each criterion?

The authors state that the maximum principal stress criterion fits the data best. From the provided figures, the maximum principal stress criterion and hayhurst criterion both look quite similar and both look like very good fits of the data. What factors influenced the authors to determine that the MPS criterion was a better fit than the hayhurst criterion?

Line 396-398: "Although the von Mises and Coulomb criteria should be disregarded on the basis…" I would recommend removing this whole sentence. Even if the results aren't as good of a fit as the others, the criteria can still provide useful information based on the assumptions they make about stress states. I would only include this bit if you weren't including all four criteria in your damage model, but since you're still using all four it seems unnecessary.

Line 416: "In contrast, peaks of $\sigma_{eq,vM}$ exceeding 200 kPa,…" I'm confused by the wording of the start of the sentence. Could it be reworded for clarity?

Line 470: "... it challenges findings based on glacier-scale in situ and remote sensing observations" These results fall squarely within the range of tensile stresses proposed by Vaughan. This would also be a part of the manuscript where I'd be interested to see some more discussion of how applicable these results are to the studies mentioned, which derive their tensile strength estimates from fast-flowing, cold ice.

Line 596-597: "we assume that damaged ice largely remains in regions where damage was initially created." It is correct that the ice remains in regions where the damage was initially created, but the stress state around them changes as the cavity fills and drains. The cavity refilling especially could create stress states that enable crevasse healing, as was observed with the stake measurements at crevasse tips which measured a closing rate of 5 mm/day. I agree that modeling crevasse healing is difficult given our current understanding of fracture mechanics and is outside the scope of this work, but this part of the reasoning should be reworded.

Figure 9: The caption states that pressure is defined as positive in compression and negative in tension, "as is standard in the glaciological literature". Pressure is defined by the principal stresses, and the standard sign convention within glaciology is that positive stresses are tensile and negative are compressive.

**Technical Corrections**

Equation 4: $\sigma_I$ should be replaced with $\sigma_{eq,mps}$
Line 82: change "on the field" to "in the field"
Line 136: change "criterion" to "criteria"
Line 151: change "framework" to "frameworks"
Line 188: change "cold ice" to "colder ice" since cold ice is often interpreted as ice colder than -10°C

Line 215: change "highest" to "largest", "eigen value" to "eigenvalue"

Line 328: change "slightly more (resp. slightly less)" to "slightly large (resp. slightly smaller)"

Line 379-380: "Only the crevasse at the upstream end of the AA' profile seem to occur…" Mismatched tense. Either "crevasse" needs to be plural or "seem" changed to "seems"

Miscellaneous figure notes:
- The magenta lines denoting unrelated crevasses are sometimes hard to see on the chosen color map. I'd recommend changing them to a color that has more contrast, or removing them from the figures entirely after introducing them in the first figure.
- Add arrows in the figures that denote the direction of ice flow
- Figure 4: if using the same color map for all displacement directions, keep color bar limits consistent across x, y, and z domains

**References**

Goldsby, D. L., and D. L. Kohlstedt. "Superplastic Deformation of Ice: Experimental Observations." *Journal of Geophysical Research: Solid Earth* 106, no. B6 (2001): 11017–30. https://doi.org/10.1029/2000JB900336.

Currier, J. H., and E. M. Schulson. "The Tensile Strength of Ice as a Function of Grain Size." *Acta Metallurgica* 30, no. 8 (August 1, 1982): 1511–14. https://doi.org/10.1016/0001-6160(82)90171-7.

---

## Referee Comment (RC2)

**Summary**

The selection of an appropriate fracture criterion for glacier ice is important for simulations of glacier hazards, including local collapse events and the larger-scale contribution of calved ice to global sea level rise. Brondex et al. make opportunistic use of a controlled cavity drainage as a natural experiment for ice damage criteria. They assert that a Hayhurst or Maximum Principal Stress criterion is the most effective for simulating crevasse initiation, while Coulomb and von Mises criteria are less effective. The authors work solely with a nonlinear viscous rheology and therefore do not directly simulate crevasse initiation, nor propagation, relying instead on a continuum damage mechanics framework to approximate the effect of crevasses. This methodological choice sets up an uncomfortable tension between fine-scale mechanics and applications to ice-sheet-scale modeling.

Overall, the manuscript is well written and organized. The mathematical frameworks in the Appendix are described well. However, the framing and motivation of some arguments is a bit muddled. The manuscript would be stronger if revised to be more focused; see General Comment 1, below.

Finally, I apologize to the authors for their long wait for review comments. The review request arrived while I was in the field and I did not see it for several weeks afterward — my fault. Thanks everyone for your patience in this process.
All the best,
Lizz Ultee

**General comments**

1. It seems that the authors want to find which failure criterion is most suitable for simulating crevasse initiation in short-term, local settings. However, they use a continuum approach, which is commonly used for large-scale ice sheet modeling, and by definition inconsistent with the initiation and propagation of crevasses. As far as I can tell, there is no explicit crevasse "initiation" here. See comments on L452, L493-496.

   • I suggest the authors explain what crevasse "initiation" means in their framework. For example, do they simulate damage evolution from 0 to non-zero values? Does this require them to "seed" (i.e. intervene to initiate) damage, or does it arise from the model?

   • I suggest that the authors choose either

     ‣ clarifying that their analysis pertains to best practices for short-term, local simulation of fracturing ice bodies using continuum methods, or

     ‣ elaborating much more on how their results might be translated to simulating "bulk calving" in large-scale ice sheet models. See comment on L505-507.

2. The authors briefly consider elastic deformation, then move on with Nye-Glen viscous simulations. I think viscoelastic simulations may be more appropriate to fit the observations they present — see comment on L341-342. However, I am not sure whether that is in line with the goals of the manuscript.

3. The authors' commentary on earlier works goes a bit beyond what their quantitative analysis can support. I suggest that the authors carefully consider the goals of their manuscript and revise accordingly.

   • The authors discuss Ultee et al 2020 — previous work of mine — for several paragraphs. While I appreciate their attention to my work, and I don't disagree with some of their points, their methods do not support a direct comparison. For example, Brondex et al.

consider linear elastic, linear viscous, and Nye-Glen viscous cases; my work mainly considered linear viscoelastic rheology, which is not treated here. Further, Brondex et al. use a more computationally expensive approach and therefore cannot sample as wide a parameter space as we did in Ultee et al 2020 (especially w.r.t. the Young's modulus and the Maxwell time). As a result, the stress thresholds Brondex et al. produce cannot be compared directly with mine. See comments on L436-437, L449-451.

> ‣ If a direct comparison is the goal, then the methods must be adjusted to support it.

> ‣ If a direct comparison is *not* among the goals of the manuscript, then I suggest the authors revise to bring their commentary in line with their goals.

• Note that similar criticism may apply to the authors' discussion of other papers; I am simply the most familiar with my own work.

**Specific comments (line by line)**

• L69-70: Possible scale problem here? The dimensions of the entire glacier (a few hundred meters wide/long) are on the order of crevasses that form on ice sheet outlet glaciers, and as the authors note later, roughly the same size as the large cavities at Skaftá. The stresses that can be produced in this setting are necessarily much lower.

• L74-82: Very cool setting for a field experiment! Nice find and nice description.

• Table 1 and throughout: typo, "Hayurst" should be "Hayhurst". Repeated in header of section 3.2.4 and elsewhere in the text.

• L199, Eqn 1: What is $Ice \sigma_{III}$? Possible failure of a subscript "Ice" somewhere?

• L215-216: It is sensible that most glacier and ice sheet studies do not consider compressive failure, because ice is much stronger in compression than in tension (Petrovic, 2003).

• L254-256: "We argue that using lower Young's modulus values to fit observations in a purely elastic model may overlook the viscous component of the observed deformation (see Sect 5.1)." — Well, sure, but this goes both ways. Using high values of the Young's modulus, as the authors do in Sect 5.1, will tend to underestimate the Maxwell time and thus overemphasize the role of viscous deformation in a material assumed to be viscoelastic.

• L280: "All pumping sequences are simulated with a one-day timestep, while the periods in between are simulated with a five-day timestep." — Here it would be helpful to address your assumption about the Maxwell time up front. One- to five-day time steps are longer than the Maxwell time you use, which motivates a simulation that includes only viscous effects on these time scales. Consider moving Figure 8 earlier?

• L302-303: Please split this "For example, …" into two sentences. The "A (resp. B) is X (rest. Y)" sentence structure is hard to parse.

• L336-340: Could you clarify what kind of uplift you see in the data? I was imagining uplift outside the edges of the cavity, as when an elastic or viscoelastic beam or plate has its center pushed down and its outer edges lift up (see e.g. Banwell et al 2013 for ice shelf case)…but I wonder if you're actually referring to uplift from the cavity refilling with water and floating the overlying ice? In the first case, it is expected that some of the stakes would be subsiding while neighboring stakes show uplift.

• L341-342: I had to read it a few times, but okay, yes, I see that the elastic displacement being two orders of magnitude too small is an argument to include viscous effects. However, I wondered about linear viscoelasticity — an initial elastic drop followed by continued viscous settling. The Nye-Glen viscous model gets the right order of magnitude deformation, but we see in Figure 7 that its velocity during the pumping operations is generally too high. I

wondered if this was a result of instantaneous stresses that immediately drive fast creep, with no elastic phase to take up the highest stresses…and if so, whether a true viscoelastic model might do a better job capturing observed cavity roof velocities.

- L389-390: "[MPS] does not offer quantitative insights, such as crevasse concentration to be expected locally. In the framework of CDM, this aspect is quantified through the damage variable D…" — A bit confusing. I typically think of damage in the vertical dimension, analogous to crevasse penetration, reaching 1 when the ice column can no longer support any load (Borstad et al., 2012). By contrast, the expected crevasse spacing in the horizontal dimension(s) is a function of the ice thickness, depth of existing crevasses, and material properties of the ice (reviewed in Colgan et al., 2016).

- L404: "…the subsequent slowdown associated with the cavity refill in late spring/summer 2011." — So the cavity roof was still subsiding (negative Vz) even during refill?

- L416-418: "In contrast, peaks of $\sigma_{eq, vM}$ …" — Rephrase for clarity? This sentence is hard to parse.

- L436-437: "[Ultee et al.'s 2020] estimate relies on the assumption that [deformation] is predominantly elastic." — I wouldn't say so. We used a linear viscoelastic model for which the best-fit Maxwell time is on the order of days, so it follows from the setup of that model that deformation was predominantly elastic. However, we also considered Glen's law displacement and reached similar conclusions (see Discussion section).

- L443-444: If you use a high Young's modulus, of course you are going to get short Maxwell times. It would be cleaner to argue here that previous authors deduced low values of Young's modulus from purely elastic fits to observations, and those fits are artificially low because they did not account for any creep. This argues for a viscoelastic framework, which is consistent with higher Young's modulus and shorter Maxwell times (Reeh et al. 2003).

- L449-451: "…we argue that Ultee et al. (2020) significantly underestimated the viscous contribution to the observed surface deformation, leading to an overestimation of stress and, as a consequence, an overestimation of ice tensile strength." — Contrary to this assertion, the parameter space we tested corresponds to Maxwell times of order 0.3-4100 hours (see Ultee et al. 2020, Table 1). We address the overestimation of stresses in our Discussion section, together with many earlier authors who have pointed out that the instantaneous elastic stresses produced in a viscoelastic model are upper bounds on what would be produced in nature (e.g. MacAyeal & Sergienko 2013). Thus, this point is not entirely relevant, and not particularly useful for your manuscript, at least as I understand it. If you wish to make this point, I suggest revising to support it better and to align it more clearly with the goals of the manuscript. See General Comments, above.

- L452: "[Our results] suggest that crevasse initiation in initially intact ice is governed by viscous processes." — This doesn't follow from the analysis you've presented. There is no crevasse "initiation" in the simulations presented.

- L462-465: It is worth questioning whether any of these methods are valid, since none include explicit crevasses, and the formation of crevasses would be assumed to concentrate (and thus "relieve") stress from surrounding areas (Rice 1968; Weertman 1973). I include my own past work in this criticism. Anyway, if you raise this point, it would be good to comment also on how it applies to your methods.

- L475-482: Consider commenting explicitly here on whether failure criteria should be based on Cauchy stress or deviatoric stress. This is a common point of confusion in ice sheet applications that I have seen in the literature. It seems you are alluding to this distinction and could communicate more clearly by naming it explicitly.

- L493-496: The notion of crevasse *initiation* being governed by viscous processes seems muddled to me. Crevasse initiation is a fundamentally discontinuous phenomenon, no? A

crevasse initiates when a discontinuity—the crevasse—forms in a medium formerly approximated as continuous.  Consider reframing for clarity.

- L505-507: In my view, most people accept that bulk calving will be implemented differently from local-scale collapse events.  Perhaps your manuscript would do best to focus more directly on the latter?

**Figures**

- Fig 1: "Surface Energy Balance station (pink triangle)" — Not seeing a pink triangle.  Is it the one nearest the cavity on panel (a)?  That one appears orange to me, the same as the other two triangle markers.

- Fig 6: Please refer back to where each equivalent stress is defined in Section 3.  I was initially very confused about why MPS was the only plot that showed compressive stresses in the center of the roof, where we are quite certain the stress should be compressive.  But in fact, this is not giving local effective stress but rather the value of the maximum principal stress, the Hayhurst stress, etc., which would be compared with the failure threshold.  Right?

- Fig 7: "Blue dots indicate mean vertical velocities derived from stake measurements, computed between two measurement dates" — Which two measurement dates? What's the temporal resolution here?

**References in this review**

Banwell, A. F., D. R. MacAyeal, and O. V. Sergienko (2013), Breakup of the Larsen B Ice Shelf triggered by chain reaction drainage of supraglacial lakes, Geophys. Res. Lett., 40, 5872–5876, doi:10.1002/2013GL057694.

Borstad, C. P., A. Khazendar, E. Larour, M. Morlighem, E. Rignot, M. P. Schodlok, and H. Seroussi (2012), A damage mechanics assessment of the Larsen B ice shelf prior to collapse: Toward a physically-based calving law, Geophys. Res. Lett., 39, L18502, doi:10.1029/2012GL053317.

Colgan, W., H. Rajaram, W. Abdalati, C. McCutchan, R. Mottram, M. S. Moussavi, and S. Grigsby (2016), Glacier crevasses: Observations, models, and mass balance implications, Rev. Geophys., 54, 119–161, doi:10.1002/2015RG000504.

Goldberg, D. N., C. Schoof, and O. V. Sergienko (2014), Stick-slip motion of an Antarctic Ice Stream: The effects of viscoelasticity, J. Geophys. Res. Earth Surf., 119, 1564–1580, doi:10.1002/2014JF003132.

MacAyeal, D. R., & Sergienko, O. V. (2013). The flexural dynamics of melting ice shelves. Annals of Glaciology, 54(63), 1–10. doi:10.3189/2013AoG63A256

Petrovic, J.J. Review Mechanical properties of ice and snow. Journal of Materials Science 38, 1–6 (2003). https://doi.org/10.1023/A:1021134128038

Reeh, N., Christensen, E. L., Mayer, C., & Olesen, O. B. (2003). Tidal bending of glaciers: a linear viscoelastic approach. Annals of Glaciology, 37, 83–89. doi:10.3189/172756403781815663

Rice, JR (1968) Mathematical analysis in the mechanics of fracture.
In Liebowitz, H ed. *Fracture: An Advanced Treatise*, Volume 2, Chapter 3. New York, USA: Academic Press, pp. 191–311.

Ultee, L., Meyer, C., & Minchew, B. (2020). Tensile strength of glacial ice deduced from observations of the 2015 eastern Skaftá cauldron collapse, Vatnajökull ice cap, Iceland. Journal of Glaciology, 66(260), 1024–1033. doi:10.1017/jog.2020.65

Weertman, J (1973) Can a water-filled crevasse reach the bottom surface of a glacier? *Proceedings of Cambridge Symposium on the Hydrology of Glaciers*, 95, pp. 139–145, International Association of Hydrological Sciences, Cambridge, UK.

---

## Author Comment (AC1)

First, we would like to thank the editor, as well as Sarah Wells-Moran and Lizz Ultee for their careful reading of our paper and for their constructive comments. We feel fortunate to have reviews from experts whose work is directly discussed in our manuscript. We respond to each review separately below. Reviewer comments are shown in black, and our responses are in blue.

**Referee #1: Sarah Wells-Moran**

**General comments**

This manuscript aims to address the question: "what is the tensile strength of ice?" using a unique set of stress data collected during the drainage of a glacier cavity. The authors utilize water pressure to directly calculate the force exerted on the glacier by slowly draining the cavity, mitigating some of the uncertainties in estimated stresses introduced by assumed rheological properties. The authors then use the finite element model Elmer/Ice to replicate the drainage with different rheological properties to see which model configuration best replicates observed stresses. The authors find that a non-linear viscous flow regime defined by n=3 best matches observations.

I found parts of the paper to be a little challenging to follow due to the structure it was presented in. Some of the readability issues are in part due to the overuse of passive voice in many sections. I have made more specific comments below on suggestions for improved readability.

Overall, I think this paper is an interesting concept with a unique dataset that approaches estimating tensile strength in a novel way, but could use further refinement in the parameters selected for the model and in the organization of the paper. My primary concern is in the choice to use a linear-viscous test case when such states have not been able to be reproduced in ice in the lab, and the absence of grain size dependent viscosity when n

Figure 1: Diagnostic effective strain rate computed using the Glen-Nye flow law (n = 3) for the glacier geometry containing an empty cavity.

First, we do not report small strain rates in the manuscript. While flow velocities at Tête Rousse are indeed much lower than those typical of Antarctic or Greenland ice streams, they vary sharply over short spatial scales in the vicinity of the cavity (see Fig. S4). These strong local velocity gradients produce elevated strain rates precisely in the region of interest, as illustrated in Fig. 1. Unfortunately, field measurements are too sparse in time and space, and only available at the surface, to reconstruct an "observed" effective strain rate. Figure 1 therefore shows the effective strain rate computed using the Glen-Nye flow law, and thus inherently assumes n=3. However, we are confident that its order of magnitude is realistic, given the demonstrated ability of the Glen-Nye law to reproduce both (i) the spatial patterns of surface deformation (Fig. 4 of our paper) and (ii) the temporal evolution of the mean vertical velocity (Fig. 7 of our paper). This good agreement between observed and modeled displacements further supports the use of n=3. This leads to our second point, namely the reviewer's statement regarding the "overestimation of the subsidence rates". This overestimation is actually minor for the 2010 drainage event and affects only the vertical component of the displacement. It can be readily explained by uncertainties in the temperature field and/or the cavity geometry. For the 2011 drainage event, the overestimation is more pronounced, but, as stated in the manuscript, this is most likely due to the collapse of ice blocks from the cavity roof, which was observed in the field but is not represented in the model. As a result, the actual roof would have been significantly lighter than the modeled one, leading to a slower observed subsidence compared to the modeled response.

More broadly speaking, the question of which value of n should be adopted in glacier-scale ice flow models is the subject of long-standing and ongoing debate (e.g. Paterson and Savage, 1963; Raymond, 1973; Thomas et al., 1980; Cuffey and Kavanaugh, 2011; Gillet-Chaulet et al., 2011; Bons et al., 2018; Millstein et al., 2022). While highly relevant, this issue lies beyond the scope of the present study. The Glen-Nye flow law remains a simplified, phenomenological relationship between stress and strain rate at the meso-scale. It does not explicitly and independently account for the diversity and complexity of the underlying deformation mechanisms occurring at the crystal scale. Upscaling these microscale processes to infer appropriate values for the flow law parameters A and n, within which this underlying complexity is embedded, remains a challenging task, and no consensus value for n has clearly emerged (e.g. Behn et al., 2021; Ranganathan and Minchew, 2024). As a result, the vast majority of modeling studies focused on glacier, ice sheet, or ice shelf dynamics adopt the traditional uniform value n = 3 (e.g., Pralong and Funk, 2005; Pattyn et al., 2008; Larour et al., 2012; Christmann et al., 2019; Cornford et al., 2020, among hundreds of others). Although several dedicated studies have explored the sensitivity of modeled ice flow to the choice of n (e.g., Zeitz et al., 2020; Getraer and Morlighem, 2025), including reviewer's own work on

the sensitivity of ice strength estimates to this parameter, this is not the focus of the present study. Instead, as emphasized in the title of our article "How to model crevasse initiation? [...]", our primary objective is to shed light on best practices for including crevasse formation in ice flow models. Accordingly, we adopt the conventional choice of n=3, which gives pretty good results in terms of deformation of the cavity roof, and do not consider that running additional simulations with n=1.8 and/or n=4, as suggested by the reviewer, would meaningfully contribute to the objectives of this work.

The majority of the tensile strength studies cited in this manuscript are derived from observations of cold ice, very fast moving ice, or both. I'd like to see some discussion about how tensile strength estimates might differ in temperate versus cold ice, and the applicability of the tensile strength estimates produced in this study to Greenland and Antarctica where the ice is flowing orders of magnitude faster and the deformation mechanism may be different. Currier and Schulson (1982) suggest tensile strength also has a grain size dependency, which may also affect the estimated tensile strength values of this study compared to estimates from Greenland or Antarctica.

First of all, we would like to stress that Tête Rousse glacier is not temperate but polythermal, as shown in Fig S1 of the Supplementary Material (see also Gilbert et al. (2012) for a detailed analysis of Tête Rousse thermal regime). Although we ran one prognostic simulation using the Glen-Nye flow law with a uniform temperature of 0°C, the reference simulation employs a non-uniform temperature field reconstructed from observations. This field reveals slightly cold ice in the main area of interest, i.e. at the surface directly above the cavity.

This being said, our study focuses on a specific configuration for which we are fortunate to have a unique and well-documented dataset, with the aim of shedding light on the best approaches to represent crevasse formation in ice flow models. This objective is clearly reflected in the title of our article, and is now explicitly stated in the Introduction. This is quite different from addressing "the question: 'what is the tensile strength of ice?'" as suggested in the reviewer's general comments. That question would imply the derivation of a universal value, or set of values, for the tensile strength of ice, applicable across all settings, something we neither claim nor attempt to achieve. On the contrary, we are careful not to overstate the generality of our tensile strength estimates, which remain grounded in a specific, well-constrained real-world event. For instance, we explicitly acknowledge that the tensile strength to be adopted in models depends on the retained failure criterion, when we state in our Conclusion that: "Adopting the usual non-linear Glen-Nye law with n=3, we show that a Maximum Principal Stress criterion, combined to a stress threshold of approximately 100 to 130 kPa, works well for modeling crevasse initiation."

That said, we believe that our findings regarding the mechanisms involved (e.g., progressive stress concentration due to the non-linearity of the Glen-Nye law, the need for a pressure-dependent failure criterion) have broader relevance beyond this particular case. In this context, we consider it both reasonable and valuable to discuss our results in relation to studies conducted in Antarctica, Greenland, Iceland, and other regions. Unfortunately, to our knowledge, no equivalent studies have been conducted on small alpine glaciers to date, although the dataset used by Vaughan (1993) contains a few mountain glaciers. Importantly, the core of our discussion does not center on differences in the absolute values of tensile strength, which we acknowledge are sensitive to parameters like grain size, deformation rate, or considered failure criterion, but rather on three broader modeling implications:

- (i) that crevasse formation (i.e. switching from a state of virgin ice to a state where crevasses start to appear) is not driven by the instantaneous elastic response to the perturbation in the force balance, but rather by the delayed viscous response that feeds back on the effective viscosity. This non-linearity thus induces higher and more concentrated stresses beyond the Maxwell time than before. Section 5.1 has been rewritten to clarify this mechanism, which is also discussed in detail in our response to the General Comments of Reviewer 2.;
- (ii) that pressure-independent failure criteria such as von Mises may lead to models predicting crevasses in areas where isotropic pressure actually prevents crevasse opening (Sect. 5.2); and
- (iii) that modeling crevasse propagation requires an additional ingredient not included in our approach (Sect. 5.3).

  In the revised version of the manuscript, we have reformulated several sentences that tended to evergeneralize

In the revised version of the manuscript, we have reformulated several sentences that tended to overgeneralize the tensile strength estimates derived from our study. We have also added a short discussion to clarify that, in contrast to the three points listed above which we believe are generally valid across mountain glaciers and ice sheets, the tensile strength values appropriate for ice flow modeling may vary slightly depending on the specific setting. This variability reflects the dependence of this material property on factors such as grain size, deformation rate, and temperature.

Many sections of this paper alternate between passive (e.g. simulations were conducted, the data were extruded) and active voice (e.g. we conduct simulations, we extrude the data), making these sections challenging to follow. I would recommend the authors switch instances of passive voice into active voice. Section 2 is especially difficult

to follow given the use of passive voice, as it is challenging to determine which actions were performed by the cited studies and which actions the authors took. Section 3 is also heavy on passive voice.

We have made a concerted effort to improve readability by replacing passive voice with active voice wherever it was appropriate and clear to do so. However, as outlined in Section 2, the Tête Rousse Glacier has been monitored since 2007 and has been the subject of numerous studies on various topics (Vincent et al., 2010; Legchenko et al., 2011; Gagliardini et al., 2011; Vincent et al., 2012; Gilbert et al., 2012; Legchenko et al., 2014; Vincent et al., 2015; Garambois et al., 2016). The dataset used in this study was compiled over multiple field campaigns spanning several years. While one or more of the co-authors were involved in many of these campaigns, this was not always the case. For instance, the drainage of the cavity was conducted not by our team, but by a private company. As a result, the use of passive voice remains appropriate in several instances, particularly where the specific actors involved are not among the co-authors.

Lines 21-23: "In addition, crevasses are involved in two processes that control the rate at which ice sheets lose mass into the ocean: iceberg calving (Colgan et al., 2016) and accelerated flow due to the softening of shear margins (Borstad et al., 2016)." This line could use more citations for each case

Regarding the softening of shear margins, we have added Sun and Gudmundsson (2023) and Surawy-Stepney et al. (2023) to the citation list. Regarding iceberg calving, Colgan et al. (2016) is a review article that already includes numerous relevant references.

Line 33-34: add Chudley et al., 2021 to citation list

**Done.**

**Section 3:**

- I would suggest a small restructuring of sections 3.2 and 3.3 to improve readability of the methods section. Rather than having the yield criteria in their own section breaking up the two sections on modeling, it may read smoother to split the last paragraph of section 3.3.3 into a new subsection solely about modeling damage, and move the descriptions of the yield criteria to the new subsection. So the flow of section 3 would be: model description → diagnostic and prognostic → implementing damage into model.
  - Section 3.1 presents the model, while Section 3.3 describes the experiments, which are two distinct topics. Section 3.2 focuses on the failure criteria considered, which are central to our study and, in our view, warrant a dedicated subsection. Furthermore, these criteria are not strictly part of the model, as the equivalent stresses associated with each criterion are computed a posteriori from the model outputs. They are not part of the experimental setup either, since the experiments are specifically designed to evaluate these criteria against observations. For these reasons, we believe the current structure of Section 3 is more appropriate than the alternative suggested by the reviewer. That said, we are happy to follow the editor's preference on this matter.
- Having the equations in the appendix made this section challenging to follow. I'd recommend integrating some of the key equations such as the Glen-Nye law, the Arrhenius relation, and the equation used to calculate pressure from the water drainage into the main text
  - Here too, we respectfully disagree with the reviewer's suggestion. In our view, the manuscript is already quite long, and the set of equations involved is very classical. These equations are generally presented in a similar manner across most ice flow modeling studies, particularly those using Elmer/Ice (e.g., Durand et al., 2009; Favier et al., 2012; Gagliardini et al., 2013; Gilbert et al., 2014; Brondex et al., 2020). We expect that most readers will already be familiar with them, and for those who are not, the Appendix provides easy access to the full formulation. Including only a subset of these equations in the main text would feel somewhat arbitrary (for example, why include the Glen-Nye law and the Arrhenius relation, but not Hooke's law or the damage evolution law?). For these reasons, we would prefer to keep the current structure. That said, we are happy to follow the editor's guidance on this point.

Section 3.2 intro: State a brief definition of equivalent stress

We have added the following sentence: "These criteria differ in how they define the equivalent stress, that is, a scalar quantity that serves as a surrogate for the complex three-dimensional stress state and that can be compared

to a scalar failure threshold."

Line 195: State the sign convention for principal stresses (e.g. tension positive compression negative)

Done.

Equation 1: define what  $Ice \sigma_{III}$  is

Thank you for pointing this out. The *Ice* was a typo. It is now corrected.

Lines 281-282: "Water pressure is applied in the cavity, i.e. at nodes identified as being above the bedrock." Wording is confusing, as all nodes in the model domain are above the bedrock. Could be reworded as "Water pressure is applied in the cavity, i.e. at nodes on the basal boundary which are not in contact with the bedrock"

**Done.**

Section 4.4: The authors detail how they set up the prognostic model with the damage-dependent viscosity and talk about where damage develops with each given model, but do not show these results in data or figures. Is it possible to include a figure showing the evolution of D over time with each criterion?

Producing damage maps at several time intervals for all four failure criteria and for all eight combinations of parameters  $(B, \sigma_{\rm th})$  would be overwhelming. Given the very low velocities (less than 1 m, a-1) and the relatively short simulation period (3 years), damage essentially remains localized where it is initially created, i.e. in regions where the equivalent stress associated with the considered failure criterion exceeds the considered stress threshold. As a result, we believe that including such maps in the main text would add limited insight. However, we have added representative damage maps to the Supplementary Material for completeness.

The authors state that the maximum principal stress criterion fits the data best. From the provided figures, the maximum principal stress criterion and hayhurst criterion both look quite similar and both look like very good fits of the data. What factors influenced the authors to determine that the MPS criterion was a better fit than the hayhurst criterion?

This is because the Hayhurst equivalent stress is calculated as:

$$\sigma_{\text{eq.H}} = \alpha \sigma_{\text{I}} + \beta \sigma_{\text{eq.vM}} + (1 - \alpha - \beta) \operatorname{tr}(\boldsymbol{\sigma}).$$
 (1)

As this expression shows,  $\sigma_{\rm eq,H}$  is a weighted combination of the maximum principal stress, the von Mises equivalent stress, and the trace of the Cauchy stress tensor. Because the upper surface is a free surface (i.e., subject to a stress-free boundary condition), one of the three principal stress is associated to the normal to the surface and is systematically zero, although numerical discretization may introduce slight deviations from zero. In crevassed regions,  $\sigma_{\rm I}$  is high (i.e.,  $\sigma_{\rm I} > 90$  kPa), and the other nonzero principal stress is of second order in the calculation of both  $\sigma_{\rm eq,vM}$  and  ${\rm tr}(\boldsymbol{\sigma})$ , as it can be seen in Figs. 6 and 9b (remembering that with our convention  $p = -{\rm tr}(\boldsymbol{\sigma})/3$ ). Concretely, in these regions (and only in these regions), we find that  $\sigma_{\rm eq,vM} \approx \sigma_{\rm I}$  and  ${\rm tr}(\boldsymbol{\sigma}) \approx \sigma_{\rm I}$ , which leads to  $\sigma_{\rm eq,H} \approx \sigma_{\rm I}$  (first and third rows of Fig. 6). This explains the similarity between the patterns obtained with the MPS and Hayhurst criteria in the crevassed areas and highlights that  $\sigma_{\rm I}$  alone is the relevant quantity for predicting crevasse formation in this case. In the revised version of the manuscript, we have rewritten Sect. 4.3 to make this explanation clearer.

Line 396-398: "Although the von Mises and Coulomb criteria should be disregarded on the basis..." I would recommend removing this whole sentence. Even if the results aren't as good of a fit as the others, the criteria can still provide useful information based on the assumptions they make about stress states. I would only include this bit if you weren't including all four criteria in your damage model, but since you're still using all four it seems unnecessary.

**Done.**

Line 416: "In contrast, peaks of  $\sigma_{eq,vM}$  exceeding 200 kPa,..." I'm confused by the wording of the start of the sentence. Could it be reworded for clarity?

We have replaced the sentence "In contrast, peaks of  $\sigma_{\rm eq,vM}$  exceeding 200 kPa, all simulations relying on the von Mises criterion produce damage on the cavity roof,..." by "In contrast, because  $\sigma_{\rm eq,vM}$  exceeds 200 kPa over most of the cavity roof, all simulations relying on the von Mises criterion produce damage in this region, ..."

Line 470: "... it challenges findings based on glacier-scale in situ and remote sensing observations" These results fall squarely within the range of tensile stresses proposed by Vaughan. This would also be a part of the manuscript where I'd be interested to see some more discussion of how applicable these results are to the studies mentioned, which derive their tensile strength estimates from fast-flowing, cold ice.

As discussed above, and as the title of the subsection implies ("5.2 What failure criterion?"), we are not referring to the values of the tensile strength estimates here, but about the choice of the failure criterion to be used. We believe this is made clear by the three sentences that precede the one you are referring to.

Line 596-597: "we assume that damaged ice largely remains in regions where damage was initially created." It is correct that the ice remains in regions where the damage was initially created, but the stress state around them changes as the cavity fills and drains. The cavity refilling especially could create stress states that enable crevasse healing, as was observed with the stake measurements at crevasse tips which measured a closing rate of 5 mm/day. I agree that modeling crevasse healing is difficult given our current understanding of fracture mechanics and is outside the scope of this work, but this part of the reasoning should be reworded."

You are right. We have removed the second reason previously given for not including damage healing. The sentence now reads: "[...] but it is not considered here, as very little information is available in the literature on the dynamics of healing in ice (Colgan et al., 2016)."

Figure 9: The caption states that pressure is defined as positive in compression and negative in tension, "as is standard in the glaciological literature". Pressure is defined by the principal stresses, and the standard sign convention within glaciology is that positive stresses are tensile and negative are compressive.

It is correct that the standard sign convention within glaciology is that positive stresses are tensile and negative are compressive. Yet, pressure is very often defined as the *opposite* of the mean normal stress. For example, in the reference textbook by Cuffey and Paterson (2010), it is written (p. 52): "The term pressure (denoted P) usually indicates the value of  $\sigma_M$  defined as positive for compression; hence  $P = -\sigma_M$ .", where  $\sigma_M = \text{tr}(\boldsymbol{\sigma})/3$  is the mean normal stress.

That said, we do not consider this to be a particularly meaningful debate, and have therefore removed the mention "as is standard in the glaciological literature" from the manuscript. However, we would prefer to retain this sign convention, as it is almost systematically adopted in studies using Elmer/Ice (e.g., Durand et al., 2009; Favier et al., 2012; Gagliardini et al., 2013; Brondex et al., 2020), where the Cauchy stress tensor  $\sigma$  is decomposed into its deviatoric part  $\bf S$  and isotropic part as:

$$\sigma = \mathbf{S} - p\mathbf{I},\tag{2}$$

which implies that pressure is defined as  $p = -\operatorname{tr}(\boldsymbol{\sigma})/3$ . We would not want to confuse readers who are familiar with this convention. Note that we have also reverted the colorbar of Fig. 9 so that blue consistently denotes compression and red denotes tension across all figures.

**Technical Corrections**

Equation 4:  $\sigma_{\rm I}$  should be replaced with  $\sigma_{\rm eq,MPS}$

Actually, no. In our approach, the Hayhurst equivalent stress is calculated considering potential negative values of the maximum principal stress  $\sigma_{\rm I}$ , as done by many authors using the Hayhurst criterion (e.g., Pralong and Funk, 2005; Jouvet et al., 2011; Duddu and Waisman, 2012; Jiménez et al., 2017; Mercenier et al., 2018, 2019). For the MPS criterion, we only consider the positive values of  $\sigma_{\rm I}$ , and thus define  $\sigma_{\rm eq,MPS} = \max(\sigma_{\rm I}, 0)$ . We are aware that other authors, including yourself, have chosen to consider only positive values of  $\sigma_{\rm I}$  when calculating the Hayhurst equivalent stress (e.g., Mobasher et al., 2016; Huth et al., 2021; Huth et al., 2023), but that is not the choice we made.

As explained above, because one of the principal stress is systematically zero at the surface due to the stressfree BC, the maximum principal stress at the surface is either positive or zero, so both choices lead to the exact same result at the surface. At depth, where the three principal stresses can be negative, we do not believe this choice to be of major importance, since the main purpose of taking only positive values of  $\sigma_{\rm I}$  is to prevent damage formation when all principal stresses are compressive. In such situations, taking  $\sigma_{\rm I}$  instead of  $\sigma_{\rm eq,MPS}$  in the calculation of  $\sigma_{\rm eq,H}$  would systematically lead to lower values of the latter, and therefore to no damage creation in any case.

```
Line 82: change "on the field" to "in the field" Line 136: change "criterion" to "criteria" Line 151: change "framework" to "frameworks"
```

The three points listed above have been corrected.

Line 188: change "cold ice" to "colder ice" since cold ice is often interpreted as ice colder than  $-10^{\circ}C$

In our view, ice is considered temperate when it is at the pressure melting point, and cold otherwise. Cold ice sticks to the bed, while temperate ice slides on the bed. We believe that, in the context of our study, the current wording does not create ambiguity and have therefore chosen to retain it.

```
Line 215: change "highest" to "largest", "eigen value" to "eigenvalue"
Line 328: change "slightly more (resp. slightly less)" to "slightly large (resp. slightly smaller)"
Line 379-380: "Only the crevasse at the upstream end of the AA' profile seem to occur..." Mismatched tense. Either "crevasse" needs to be plural or "seem" changed to "seems"
```

The three points listed above have been corrected.

**Miscellaneous figure notes:**

- The magenta lines denoting unrelated crevasses are sometimes hard to see on the chosen color map. I'd recommend changing them to a color that has more contrast, or removing them from the figures entirely after introducing them in the first figure.

We have tried many colors but could not find one that provides good contrast with all the colormaps used in the main text and Supplementary Material figures. We have decided to remove them from all figures except for Fig. 1.

- Add arrows in the figures that denote the direction of ice flow
  - We have added arrows indicating the ice-flow direction in Fig. 1 as well as in Figs. 4, S5, and S6. We consider it unnecessary and visually overloading to add such arrows in the other figures, as they share the same orientation as Fig. 1.
- Figure 4: if using the same color map for all displacement directions, keep color bar limits consistent across x, y, and z domains.

We have modified the figure so that the same colorbar limits are now used for the x- and y-directions. We kept a separate color scale for the z-direction because adopting a single scale that spans the full range of displacement magnitudes in both the horizontal and vertical directions would result in insufficient resolution for the x- and y-direction displacements, making meaningful comparison with the observations difficult. We believe that displaying the color scale alongside each map provides sufficient clarity. To further emphasize this, we have slightly revised the figure caption, which now contains: "Note that displacement units are in millimeters for the elastic case and in centimeters for the other cases, and that the colorbar scale is not the same across the horizontal and vertical directions."

**Referee #2: Lizz Ultee**

**Summary**

The selection of an appropriate fracture criterion for glacier ice is important for simulations of glacier hazards, including local collapse events and the larger-scale contribution of calved ice to global sea level rise. Brondex et al. make opportunistic use of a controlled cavity drainage as a natural experiment for ice damage criteria. They assert that a Hayhurst or Maximum Principal Stress criterion is the most effective for simulating crevasse initiation, while Coulomb and von Mises criteria are less effective. The authors work solely with a nonlinear viscous rheology and therefore do not directly simulate crevasse initiation, nor propagation, relying instead on a continuum damage mechanics framework to approximate the effect of crevasses. This methodological choice sets up an uncomfortable tension between fine-scale mechanics and applications to ice-sheet-scale modeling.

Overall, the manuscript is well written and organized. The mathematical frameworks in the Appendix are described well. However, the framing and motivation of some arguments is a bit muddled. The manuscript would be stronger if revised to be more focused; see General Comment 1, below.

Finally, I apologize to the authors for their long wait for review comments. The review request arrived while I was in the field and I did not see it for several weeks afterward - my fault. Thanks everyone for your patience in this process.

All the best, Lizz Ultee

**General comments**

- 1. It seems that the authors want to find which failure criterion is most suitable for simulating crevasse initiation in short-term, local settings. However, they use a continuum approach, which is commonly used for large-scale ice sheet modeling, and by definition inconsistent with the initiation and propagation of crevasses. As far as I can tell, there is no explicit crevasse "initiation" here. See comments on L452, L493-496.
  - I suggest the authors explain what crevasse "initiation" means in their framework. For example, do they simulate damage evolution from 0 to non-zero values? Does this require them to "seed" (i.e. intervene to initiate) damage, or does it arise from the model?
  - I suggest that the authors choose either
    - clarifying that their analysis pertains to best practices for short-term, local simulation of fracturing ice bodies using continuum methods, or
    - elaborating much more on how their results might be translated to simulating "bulk calving" in large-scale ice sheet models. See comment on L505-507.

We confirm that we use a continuum approach, as reflected in the governing equations solved by our model. This implies that individual crevasses are not explicitly represented in the model: the ice geometry remains continuous throughout the simulations. Although our study focuses on a small glacier allowing finer mesh resolutions than those typically used in large-scale ice-sheet simulations, this does not change the fundamental nature of the approach. We do not consider the distinction between "short-term, local settings" and "large-scale ice sheet modeling" to be relevant in this regard.

Our objective is to investigate under which stress states and magnitudes an initially crevasse-free region becomes crevassed. This is what we mean by "crevasse initiation". This is conceptually similar, aside from the additional failure criterion aspect, to the definition proposed in the reviewer's own work, where it is stated that "we refer to ice strength as a bulk property equaling the maximum stress intact ice can sustain before fractures appear at the macroscale" (Ultee et al., 2020). We have revised parts of the manuscript to clarify this point. From a Continuum Damage Mechanics (CDM) perspective, this is indeed equivalent to constrain the source term of the damage evolution equation, i.e. the stress state and magnitude beyond which an undamaged area (D = 0) begins to accumulate damage (D > 0). This does not require to seed damage. Damage can emerge from a virgin state (D = 0) when the chosen failure criterion is met, i.e. when  $\sigma_{eq}$  exceeds the stress threshold as shown by equations (A10) and (A11).

As discussed in Section 5.3, the fact that individual crevasses are not explicitly represented in the model means that stress concentrations at crack tips cannot be accounted for. Beyond a critical crevasse depth, these stress

concentrations become the dominant factor governing fracture propagation. It follows that our approach alone is not sufficient to simulate ice-detachment processes such as calving or serac fall. This is why we argue in Section 5.3 that attempts to model calving solely within a purely viscous framework, in particular through CDM (a clarification we have now added), necessarily omit key fracture physics and therefore rely on failure criteria that lack physical justification. We conclude this section by saying that "our findings support modeling approaches that couple a viscous flow model to describe crack nucleation with an LEFM approach to simulate subsequent fracture propagation." We have reformulated some parts of this Section 5.3 to make this point clearer.

2. The authors briefly consider elastic deformation, then move on with Nye-Glen viscous simulations. I think viscoelastic simulations may be more appropriate to fit the observations they present — see comment on L341-342. However, I am not sure whether that is in line with the goals of the manuscript.

If the statement "viscoelastic simulations may be more appropriate to fit the observations" refers to the observed mean vertical velocities shown in Figure 7 as suggested by the comment on lines 341–342, this point is addressed in our reply to that specific comment.

With regard to the evolution of stresses in response to the perturbation (the main focus of our paper), the only added value of viscoelastic simulations would be to capture the transient transition between the two limiting regimes (purely elastic and purely viscous) already treated in our analysis. As explained in the following paragraph, the results presented in the submitted version of our paper are sufficient to demonstrate that such simulations are not necessary for the purpose of our study. Nevertheless, for completeness and to address this suggestion rigorously, we have now performed these simulations, which partly explains the delay in providing our response to the reviewers. The results of these new simulations are presented in two dedicated paragraphs, followed by a concluding paragraph summarizing our response.

On why viscoelastic simulations are unnecessary We disagree with the common distinction often made in the literature between an "elastic stress" and a "viscous stress". Consistent with Maxwell-type viscoelastic models, the stress field at any given time is simply the solution of the force balance. It is therefore meaningless to split the instantaneous stress into separate elastic and viscous components. What is physically meaningful is that a given stress field produces a total deformation that can be decomposed into an instantaneous elastic contribution and a delayed viscous contribution. This distinction has important consequences when using the Glen-Nye (note that we have replaced "Glen's law" by "the Glen-Nye law" everywhere in the paper) flow law because viscosity is non-linear: even in diagnostic simulations, the feedback of viscous deformation on effective viscosity occurs "instantaneously" within the model and thus directly alters the computed stress field. In nature, however, that feedback requires time to develop (the Maxwell time) because viscous strains are delayed. In this sense, the stress fields modeled with the Glen-Nye law are representative of behavior beyond the Maxwell time, whereas the purely elastic solution is representative only of very short timescales. For completeness, we note a further difference between the regimes: elastic behavior is usually modeled as compressible while ice is assumed incompressible in the purely viscous regime, which inherently leads to different stress solutions under the two assumptions. The diagnostic experiments described in the previous version of the paper already explore these two end-member mechanical behaviors. In Section 4.1, we show that even under a worst-case scenario (i.e., an instantaneous drainage of the cavity), the magnitude and spatial pattern of stresses predicted in the purely elastic regime are insufficient to explain crevasse initiation. Only after viscous deformation has fed back on viscosity do stress levels compatible with crevasse formation (i.e., those predicted with the Glen-Nye law) emerge. Indeed, this feedback causes stress redistribution and results in higher and more concentrated tensile stresses around the cavity in the delayed viscous regime than in the immediate elastic response, a result that might seem counter-intuitive if one assumes the initial elastic response always produces the largest stresses. Yet, this result is also supported by the Maxwell time estimates around the cavity, which range between about 10 hours to 1 day for a Young's modulus of E=1 GPa (former Figure S8 which is now Figure 8). If crevasses had opened within this timescale after drainage, they would likely have been noticed and reported by field operators because of safety concerns. Yet no such observations were made; crevasses were only detected the following summer after snowmelt, approximately 10 months after the drainage operation (although they likely formed earlier but remained hidden beneath snow).

Description and results of the new viscoelastic simulations. These new simulations are based on the experimental setup used for the diagnostic experiments presented in Section 3.3.2 of the paper, except that they are run in transient mode. Indeed, since the unknowns of a Maxwell viscoelastic problem are displacements that

Figure 2: Temporal evolution of surface anomalies in maximum principal stress induced by the presence of an empty cavity, modeled using a linear Maxwell viscoelastic rheology with  $\nu = 0.45$ . The upper-left panel shows the anomaly computed assuming purely elastic behavior with  $\nu = 0.3$ , while the lower-right panel shows the anomaly obtained for a purely linear viscous rheology under the assumption of ice incompressibility.

Figure 3: Same as Fig. 2, but for the non-linear (Glen-Nye) Maxwell viscoelastic rheology.

Figure 4: Surface anomalies in maximum principal stress induced by the presence of an empty cavity, modeled assuming pure elasticity with  $\nu = 0.45$  (upper left), linear viscoelasticity with  $\nu = 0.45$  (upper right), pure elasticity with  $\nu = 0.5$  (middle left), linear viscoelasticity with  $\nu = 0.5$  (middle right), and a pure linear viscous and incompressible rheology (lower right).

accumulate over time, there is no steady-state solution to this problem, unlike in the purely viscous case, for which a steady velocity field can be computed for any given geometry. Concretely, we run two sets of simulations for both the empty-cavity and cavity-free geometries: one lasting five minutes with a one-second time step, and another lasting one day with a five-minute time step. We test both a linear Maxwell viscoelastic rheology and a non-linear (Glen-Nye) Maxwell viscoelastic rheology. Note that in the latter case, only the viscous component is non-linear, while the elastic part remains linear. For both rheologies, we use a Poisson's ratio of  $\nu=0.45$ . We choose this value in line with, e.g., Gudmundsson (2011) or Rosier et al. (2014), because, as noted by MacAyeal et al. (2015) and discussed above, a viscoelastic model must reconcile two conflicting rheologies: the elastic rheology in which ice is considered compressible ( $\nu=0.3$ ), and the viscous rheology in which ice is assumed incompressible. Figures 2 and 3 show the temporal evolution of surface anomalies (i.e., the difference between the solutions obtained in the empty-cavity and no-cavity configurations) in maximum principal stress induced by the presence of an empty cavity for the linear and non-linear viscoelastic rheologies, respectively. Note that the colorbar ranges differ between the two figures.

Let us first focus on the linear case (Fig. 2). After one second of simulation, the viscoelastic solution is very close to the purely elastic one; the small differences observed in the stress pattern are entirely due to the difference in Poisson's ratio, i.e.  $\nu = 0.3$  in the elastic case versus  $\nu = 0.45$  in the viscoelastic case (see Fig. 4). Similarly, the differences between the patterns obtained with the viscoelastic and purely viscous solutions are mainly attributable to the assumption on ice compressibility ( $\nu = 0.45$  for the viscoelastic case versus full incompressibility for the purely viscous case). As shown in Fig 4, the fact that the stress pattern is controlled primarily by the choice of Poisson's ratio, rather than by the nature of the constitutive law, is clearly established by two additional simulations testing  $\nu = 0.45$  and  $\nu = 0.5$  (i.e., incompressible) for the purely elastic case, and  $\nu = 0.5$  for the linear viscoelastic case. It is striking to observe how similar the diagnostic stress field computed with the purely viscous law is to those obtained with the purely elastic or viscoelastic laws when  $\nu = 0.5$  (incompressible). This similarity arises because the "instantaneous" force equilibrium yields the same stress solution regardless of the constitutive law, provided that: (i) the geometry is identical, (ii) the assumption regarding compressibility is the same, (iii) the same forces are applied at the domain boundaries (hence the importance of the force-driven experiment concept as Dirichlet boundary conditions would imply different forces at the boundaries of the domain depending on the values attributed to the constitutive law parameters), and (iv) there is no feedback of stresses through strain or strain rate on the rheological parameters (i.e., both elasticity and viscosity are linear). This point is already discussed in the paper (beginning of Sections 4.1 and 5.1, supported by Figs. S2-S3-S4). Returning to Fig. 2, the stresses computed with the viscoelastic law gradually evolve toward higher tensile stresses around the cavity roof as deformations accumulate over time. The feedback of deformation on the stress field in the viscoelastic simulation is equivalent to the stress evolution that would result from an evolving geometry in the purely viscous simulation (we recall that the stress solution computed in the latter is a diagnostic stress field for the initial, undeformed geometry).

Now, let us consider the non-linear case (Fig. 3). Again, the viscoelastic solution obtained after 1 s of simulation is very close to the purely elastic one, with the small differences being attributable to the different values of Poisson's ratio. As in the linear case, the stresses computed with the non-linear viscoelastic law evolve over time toward higher tensile stresses around the cavity roof. However, this evolution is much stronger and occurs much faster than in the linear case (note again that the color scale differs between the two regimes), because it results not only from the accumulation of deformation, but also from the *progressive* feedback of strain rates on viscosity. This feedback makes the ice more fluid, leading to a concentration of stresses as discussed above. In this case, the solution obtained with the purely viscous (Glen-Nye) law more closely resembles the viscoelastic solution after several hours of simulation, rather than at the very beginning, as was observed for the linear case. As explained above, this is because, although the pure Glen-Nye solution does not account for geometric changes (we are showing the steady-state solution), it does capture the feedback of strain rate on viscosity. Moreover, while this feedback develops progressively in the viscoelastic case, it affects viscosity instantaneously in the purely viscous one. In the present case, we observe that after approximately 12 to 24 h of simulation, the viscoelastic solution has essentially converged toward the purely viscous one, with the remaining small differences in pattern again attributable to the different assumptions regarding ice compressibility (plus the fact that the Glen-Nye solution is a steady state while the viscoelastic one is transient and thus accumulates deformations). This result is consistent with the Maxwell time of 10 hours to 1 day in the vicinity of the cavity with E=1 GPa (Fig. 8 of the paper).

To sum up. The non-linear viscoelastic simulation confirms that the solution computed using the Glen-Nye flow law represents the actual stress state in the domain, provided that sufficient time has elapsed since the perturbation for the viscous deformations (which are delayed in time) to feed back on viscosity. This characteristic timescale is the Maxwell time. Accounting for viscoelasticity would be critical only if the system was subjected to repeated

perturbations at time intervals shorter than the Maxwell time. This is not the case here, and capturing the transition from the purely elastic to the purely viscous regime in response to cavity drainage is a big computationnal effort (very short time steps are required for viscoelastic simulations) that proves unnecessary for the purpose of our study. In fact, considering the purely viscous regime alone is sufficient for assessing crevasse onset in this case. These new simulations also confirm the crucial importance of accounting for the non-linear nature of ice viscosity, which leads to strong stress concentration, an effect without which the observed crevasse field cannot be explained. Consequently, we have decided to only shortly mention these simulations in the main paper without describing them in details, while providing a more detailed description and Figures 2 and 3 in the Supplementary Material. We have also substantially revised Section 5.1 of the paper to incorporate the explanations provided above.

- 3. The authors' commentary on earlier works goes a bit beyond what their quantitative analysis can support. I suggest that the authors carefully consider the goals of their manuscript and revise accordingly.
  - The authors discuss Ultee et al 2020 previous work of mine for several paragraphs. While I appreciate their attention to my work, and I don't disagree with some of their points, their methods do not support a direct comparison. For example, Brondex et al. consider linear elastic, linear viscous, and Nye-Glen viscous cases; my work mainly considered linear viscoelastic rheology, which is not treated here. Further, Brondex et al. use a more computationally expensive approach and therefore cannot sample as wide a parameter space as we did in Ultee et al 2020 (especially w.r.t. the Young's modulus and the Maxwell time). As a result, the stress thresholds Brondex et al. produce cannot be compared directly with mine. See comments on L436-437, L449-451.
    - If a direct comparison is the goal, then the methods must be adjusted to support it.
    - If a direct comparison is \*not\* among the goals of the manuscript, then I suggest the authors revise to bring their commentary in line with their goals.
  - Note that similar criticism may apply to the authors' discussion of other papers; I am simply the most familiar with my own work.

A direct comparison with the reviewer's work is not the goal of our study. As stated in the title, our objective is to use insights from the artificial drainage of the Tête Rousse cavity to investigate how crevasse initiation can be modeled. That being said, both the reviewer's study and ours look at the magnitude of stresses generated in a crevassed area in response to ice subsidence induced by the drainage of an englacial water reservoir, with the aim of estimating an effective tensile strength for ice. While we obtain a tensile strength on the order of  $\sim 100$  kPa, the reviewer and her colleagues report a value of about  $\sim 1$  MPa, which is an order of magnitude higher. Whatever the cause of this discrepancy (whether due to an inappropriate constitutive law, inadequate parameter values, or a difference of scale, all these points are addressed in the specific comments below), we believe that it necessarily warrants discussion.

The same reasoning applies to our discussion of other studies. In particular, we consider it relevant to propose an explanation for why, according to our simulations, a Maximum Principal Stress (MPS) or Hayhurst criterion performs better in reproducing crevasse formation than a Coulomb or von Mises criterion, whereas glacier-scale field and remote-sensing studies generally support the latter two. We have therefore decided to retain these discussions in the manuscript, although Section 5.1 has been substancially rewritten as explained above.

**Specific comments (line by line)**

L69-70: Possible scale problem here? The dimensions of the entire glacier (a few hundred meters wide/long) are on the order of crevasses that form on ice sheet outlet glaciers, and as the authors note later, roughly the same size as the large cavities at Skaftá. The stresses that can be produced in this setting are necessarily much lower.

It is true that there is a large difference in scale between our setting and the one considered in the reviewer's work, and the stresses generated during the initial phase of the Skaftá cauldron collapse may indeed have been higher than those produced in our case. However, this does not imply that the stress threshold itself should differ between the Skaftá and Tête Rousse cases. Rather, it suggests that crevasses may have initiated earlier in the process in the former than in the latter case. In the reviewer's paper, it is written that "Rings of fractures are visible at the ice surface [...] on 10 October 2015, just 7 days after the end of the drainage event." It is unfortunately not clear whether this means that crevasses were absent beforehand or that no imagery was available prior to that date. In the latter case, we would be interrested to know when is the last image with no crevasse visible. In the

former case, this would be compatible with a crevasse formation in the viscous regime (i.e. after sufficient feedback of viscous deformation on ice viscosity, see above), with the viscous regime being reach faster in the Skaftá than in the Tête Rousse case due to, indeed, higher stresses and thus strain rates. This is basically what we mean when we say in the paper that it is likely that the Maxwell time near the cauldron was shorter in Skaftá than in the Tête Rousse case.

Another point worth raising is that scale effects are commonly invoked, among other factors, to explain why tensile strengths of ice derived from laboratory experiments are typically an order of magnitude higher than those estimated from in situ observations (e.g., Grinsted et al., 2024), with the reviewer's finding of  $\sigma_{\rm th} \sim 1$  MPa being one notable exception. This difference has been attributed to the way the probability of weakest links scales with sample volume (Dempsey et al., 1999; Petrovic, 2003). However, if that kind of scale effects were the explanation for the discrepancy between our results and those of the reviewer, the trend should be reversed: we would expect to find a higher tensile strength than the reviewer and her colleagues, not a lower one.

As discussed in detail below, we believe that the discrepancy between the tensile strength estimated by the reviewer and her colleagues and our estimate is not a scale effect, but rather results from an overestimation of stresses caused by an underestimation of delayed viscous deformation and its feedback on viscosity in their case.

L74-82: Very cool setting for a field experiment! Nice find and nice description.

**Thank you.**

Table 1 and throughout: typo, "Hayurst" should be "Hayhurst". Repeated in header of section 3.2.4 and elsewhere in the text.

Thank you for noticing. It has been corrected.

L199, Eqn 1: What is  $Ice\sigma_{III}$ ? Possible failure of a subscript "Ice" somewhere?

It was a typo. It has been corrected. Thank you for noticing.

L215-216: It is sensible that most glacier and ice sheet studies do not consider compressive failure, because ice is much stronger in compression than in tension (Petrovic, 2003).

We totally agree on that point.

L254-256: "We argue that using lower Young's modulus values to fit observations in a purely elastic model may overlook the viscous component of the observed deformation (see Sect 5.1)." - Well, sure, but this goes both ways. Using high values of the Young's modulus, as the authors do in Sect 5.1, will tend to underestimate the Maxwell time and thus overemphasize the role of viscous deformation in a material assumed to be viscoelastic.

That is, of course, true. What we meant here is that Young's modulus is a material property, not a tuning parameter, and therefore cannot take arbitrary values. Laboratory measurements based on the propagation of sound waves yield values around E=9 GPa (Cuffey and Paterson, 2010). Studies investigating ice-shelf flexure in response to tides often adopt values of approximately E=1 GPa (e.g., Vaughan, 1995; Sykes et al., 2009; Gudmundsson, 2011), which is already an order of magnitude lower than laboratory-derived estimates. We therefore consider that remaining within the range E=1–9 GPa is sound. That said, we have now switched Figures 8 and S8: the Maxwell time figure for E=1 GPa is included in the main text, while the one for E=9 GPa has been moved to the Supplementary Material.

L280: "All pumping sequences are simulated with a one-day timestep, while the periods in between are simulated with a five-day timestep." - Here it would be helpful to address your assumption about the Maxwell time up front. One- to five-day time steps are longer than the Maxwell time you use, which motivates a simulation that includes only viscous effects on these time scales. Consider moving Figure 8 earlier?

Yes, that is correct. As shown by the non-linear viscoelastic simulations presented in the General Comments, the stresses computed with the viscoelastic rheology become similar to those obtained with the Glen-Nye law after approximately 12–24 h of simulation. This similarity supports the relevance of the one- to five-day time steps used

in the purely viscous (Glen-Nye) simulations. This is especially true considering that the results presented above correspond to a worst-case scenario (instantaneous drainage), whereas the transient simulations presented in the paper account for the slow drainage of the cavity. We prefer to keep the discussion of the Maxwell time (and, consequently, Figure 8) in the Discussion section, but we now refer to the Maxwell time to justify our choice of time-step size.

L302-303: Please split this "For example,..." into two sentences. The "A (resp. B) is X (rest. Y)" sentence structure is hard to parse.

**Done.**

L336-340: Could you clarify what kind of uplift you see in the data? I was imagining uplift outside the edges of the cavity, as when an elastic or viscoelastic beam or plate has its center pushed down and its outer edges lift up (see e.g. Banwell et al 2013 for ice shelf case)...but I wonder if you're actually referring to uplift from the cavity refilling with water and floating the overlying ice? In the first case, it is expected that some of the stakes would be subsiding while neighboring stakes show uplift.

The uplift we refer to is clearly visible in Fig. S5 of the Supplementary Material, particularly in the bottom-right panel showing observed vertical motion. It is also apparent in Fig. 7 of the main paper, where a few data points correspond to observed mean vertical velocities above the 0 mm/d line during late summer to early autumn 2011. As noted in the manuscript (L329–330), this period corresponds to when the cavity was filled with water and under pressure, i.e., the water was pushing against the cavity roof. We have reformulated the sentence in the manuscript to clarify this point.

L341-342: I had to read it a few times, but okay, yes, I see that the elastic displacement being two orders of magnitude too small is an argument to include viscous effects. However, I wondered about linear viscoelasticity — an initial elastic drop followed by continued viscous settling. The Nye-Glen viscous model gets the right order of magnitude deformation, but we see in Figure 7 that its velocity during the pumping operations is generally too high. I wondered if this was a result of instantaneous stresses that immediately drive fast creep, with no elastic phase to take up the highest stresses... and if so, whether a true viscoelastic model might do a better job capturing observed cavity roof velocities.

First of all, we emphasize that the mean vertical velocity simulated with the Glen-Nye flow law during the 2010 drainage event is consistent with field measurements. Only the 2011 drainage sequence shows a clear overestimation of the subsidence rate by the model. As stated in the manuscript, "This could be attributed to uncertainties regarding the cavity shape that increase over time, particularly with the detachment of ice blocks from the roof. This detachment, observed in the field but not accounted for in the model, would have lightened the roof of the cavity, thereby reducing the gravitational forces compared to those estimated in the model."

Beyond this point, Figure 8 of the paper, together with the results of the non-linear viscoelastic simulation presented above, clearly show that, after only a few hours (one day at most) following the perturbation, the purely viscous regime is reached and elastic effects can be neglected. This timescale is much shorter than the time-averaging windows used to calculate the vertical velocities shown in Figure 7: there was at least a four-day interval between two measurements of the stake positions (see response to the specific comment on Fig. 7). In other words, the transition between the purely elastic and purely viscous regimes occurs over a timescale that is well below the temporal resolution of the velocity measurements. Therefore, it is consistent to compare the observed velocities with those obtained from a pure Glen-Nye viscous law using time steps of one to five days, even at the beginning of the drainage operation.

L389-390: "[MPS] does not offer quantitative insights, such as crevasse concentration to be expected locally. In the framework of CDM, this aspect is quantified through the damage variable D..." - A bit confusing. I typically think of damage in the vertical dimension, analogous to crevasse penetration, reaching 1 when the ice column can no longer support any load (Borstad et al., 2012). By contrast, the expected crevasse spacing in the horizontal dimension(s) is a function of the ice thickness, depth of existing crevasses, and material properties of the ice (reviewed in Colgan et al., 2016).

As discussed in Section 5.3 and in the General Comments, continuum approaches such as CDM cannot represent

crevasse propagation. Consequently, the vertical extent of the damaged zone cannot be directly interpreted as the crevasse penetration depth, although some authors use an arbitrary damage threshold to define an initial fracture depth for a LEFM model that subsequently governs whether the fracture propagates deeper or not (Krug et al., 2014). As stated in our manuscript, "Continuum Damage Mechanics (CDM) can be used to account for the degradation of ice mechanical properties at the mesoscale caused by the nucleation of cracks that are too small to be explicitly resolved in the continuum model. This is achieved by prescribing a dependence of viscosity on a physical state variable known as damage." (L.166-169). The magnitude of the effect of damage on viscosity depends directly on the value taken by the damage variable through Eqs. (A7)-(A9). What we meant in the sentence referred to by the reviewer is that the stress threshold and the chosen failure criterion govern only the "binary information", that is whether or not damage should be produced locally. If it should be produced, the rate at which it is produced is controlled by another parameter of the CDM model, namely the damage enhancement factor B. For completeness, note that the magnitude of the difference between the local equivalent stress and the stress threshold also influences the rate of damage production. We have reformulated Section 4.4 in the revised manuscript to make these points clearer.

L404: "...the subsequent slowdown associated with the cavity refill in late spring/summer 2011." - So the cavity roof was still subsiding (negative Vz) even during refill?

Yes. This is clearly shown in Fig. 7, where both observed and modeled vertical velocities are almost always negative. The short periods of positive vertical velocities observed in September 2011 indicate that the cavity was under pressure (water pressure exceeding the ice overburden pressure), which prompted rapid drainage for safety reasons. For the remainder of the time, the cavity is not full, and its roof is therefore not in contact with water. As a result, it continues to subside even during the refilling stages, although changes in the force balance around the cavity walls due to the rising water level have repercussions throughout the cavity and lead to a slowdown in the subsidence.

L416-418: "In contrast, peaks of  $\sigma_{\rm eq,vM}$  ..." - Rephrase for clarity? This sentence is hard to parse.

Reviewer 1 had the same remark. We have therefore reformulated as follows: "In contrast, because  $\sigma_{\rm eq,vM}$  exceeds 200 kPa over most of the cavity roof, all simulations relying on the von Mises criterion produce damage in this region, ..."

L436-437: "[Ultee et al.'s 2020] estimate relies on the assumption that [deformation] is predominantly elastic." - I wouldn't say so. We used a linear viscoelastic model for which the best-fit Maxwell time is on the order of days, so it follows from the setup of that model that deformation was predominantly elastic. However, we also considered the Glen-Nye law displacement and reached similar conclusions (see Discussion section).

**See answer below.**

L443-444: If you use a high Young's modulus, of course you are going to get short Maxwell times. It would be cleaner to argue here that previous authors deduced low values of Young's modulus from purely elastic fits to observations, and those fits are artificially low because they did not account for any creep. This argues for a viscoelastic framework, which is consistent with higher Young's modulus and shorter Maxwell times (Reeh et al. 2003).

**See answer below.**

L449-451: "...we argue that Ultee et al. (2020) significantly underestimated the viscous contribution to the observed surface deformation, leading to an overestimation of stress and, as a consequence, an overestimation of ice tensile strength." - Contrary to this assertion, the parameter space we tested corresponds to Maxwell times of order 0.3-4100 hours (see Ultee et al. 2020, Table 1). We address the overestimation of stresses in our Discussion section, together with many earlier authors who have pointed out that the instantaneous elastic stresses produced in a viscoelastic model are upper bounds on what would be produced in nature (e.g. MacAyeal & Sergienko 2013). Thus, this point is not entirely relevant, and not particularly useful for your manuscript, at least as I understand it. If you wish to make this point, I suggest revising to support it better and to align it more clearly with the goals of the manuscript. See General Comments, above.

The three points listed above are addressed together here. First, to ensure that we are aligned, we summarize our understanding of the reviewer's and her colleagues' approach (please correct us if we have misunderstood).

The first approach, described in Section 3.1 of Ultee et al. (2020), estimates the stress required to reproduce the observed deformation assuming that the deformation is entirely elastic (no force balance equation here). As acknowledged by the authors in their Discussion section, this approach clearly overestimates elastic deformation and therefore the stresses.

The second approach approximates the cauldron as a circular plate and solves the force balance for this plate (Eq. 13), assuming a linear viscoelastic rheology. The authors justify the use of a linear rather than a non-linear viscosity as follows: "Previous authors studying timescales between the Maxwell time and the long-timescale viscous limit have allowed non-linear, Glen's law creep in the viscosity η (Goldberg and others, 2014; Robel and others, 2017). Here, by contrast, we study the cauldron system close to the Maxwell time  $t_r$ , such that the response to forcing is predominantly elastic with viscous deformation becoming apparent only later. For this reason, we take viscosity  $\eta$  to be linear.". From this justification, it seems that the statement that "the response is predominantly elastic" is a modeling assumption rather than a modeling result. Using this linear viscoelastic framework, the authors ran several simulations sampling the parameter space listed in their Table 1. We note that this parameter space includes geometrical variables, and notably the ice thickness above the cauldron that directly affects the load applied to the circular plate. In contrast, in our case the cavity geometry is well constrained and we don't need to "sample the parameter space" as suggested by the reviewer in her general comments. In Figure 6 of their paper, the authors present modeled deformations (instantaneous elastic, viscous after two days, and viscous after four days) alongside the observed glacier surface on 10 October 2015 (seven days after the end of the drainage event). It is not explicitly stated which parameter set these results correspond to, but we assume they correspond to the "best-fit" parameters that provide the best match to observations and that are listed as "representative values" in Table 1. If this is the case, the "best-fit" Maxwell time is approximately  $\sim 11$  h rather than "of the order of days". If not, we would appreciate clarification on whether the "best-fit" parameters differ from those listed as "representative values".

Anyway, we argue that using linear viscoelasticity instead of a non-linear (Glen-Nye) viscoelasticity leads to an underestimation of the viscous contribution to the observed surface deformation. This is because the strong feedback between strain rate and effective viscosity in the Glen-Nye law strongly enhances ice flow over the cauldron roof. In the reviewer's Figure 6, it is striking that the viscous displacement during the first two days (on top of the initial elastic deformation) is of similar magnitude to that produced during the following two days (between day 2 and day 4). With a non-linear viscosity, one would instead expect greater displacement between day 2 and day 4 due to viscosity reduction through increasing strain rates. The consequence of underestimating viscous deformation is that the model must overestimate elastic deformation somehow to match observations. In Figure 6, it is also evident that four days of viscous deformation produce a much smaller displacement than the initial elastic one, which appears inconsistent with a Maxwell time of approximately 11h. Concretely, this overestimated elastic deformation arises from the chosen parameter set. In particular, if the Young's modulus and cauldron radius are assumed to be known, obtaining larger elastic displacements to compensate for the insufficient viscous ones requires increasing the applied load, which can only be achieved by "exaggerating" the cauldron's ice thickness, leading in turn to overestimated stresses.

To summarize, as stated in our general comments, our goal is not to directly compare our work to that of the reviewer, but rather to provide a possible explanation for why our estimated tensile strength is roughly an order of magnitude lower. We argue that this difference likely stems from an overestimation of stresses in the reviewer's study related to an underestimation of viscous deformation in both the purely elastic and linear viscoelastic approaches. We believe this explanation is relevant for the reader and helps reconcile the two studies.

L452: "[Our results] suggest that crevasse initiation in initially intact ice is governed by viscous processes." - This doesn't follow from the analysis you've presented. There is no crevasse "initiation" in the simulations presented.

This point is addressed in the general comments.

L462-465: It is worth questioning whether any of these methods are valid, since none include explicit crevasses, and the formation of crevasses would be assumed to concentrate (and thus "relieve") stress from surrounding areas (Rice 1968; Weertman 1973). I include my own past work in this criticism. Anyway, if you raise this point, it would be good to comment also on how it applies to your methods.

We generally agree with this point. However, because observational methods derive strain-rate fields from observed surface velocities, the resulting stress fields are inevitably affected by the spatial and temporal resolution of the available data anyway (Grinsted et al., 2024). It is not certain that the uncertainty in the very local stress field

around crevasses stemming from the neglected stress concentrations is systematically larger than that introduced by the averaging inherent to these observational approaches. Regarding our own work, we have added a sentence acknowledging this effect, while noting that "as no crevasses were reported during the 2010 drainage operations, it is unlikely that the stress fields computed during this sequence are affected by this limitation.".

L475-482: Consider commenting explicitly here on whether failure criteria should be based on Cauchy stress or deviatoric stress. This is a common point of confusion in ice sheet applications that I have seen in the literature. It seems you are alluding to this distinction and could communicate more clearly by naming it explicitly.

We consider it is better to stick to the notion of pressure-dependent versus pressure-independent criteria, although both notions are related. This is because a criterion can be written in terms of the Cauchy stresses and still be pressure independent (e.g., von Mises criterion as expressed in our Eq. (2)). Of course, pressure being defined as  $p = -\operatorname{tr}(\boldsymbol{\sigma})/3$  (with our sign convention, see responses to reviewer 1), pressure-dependent criteria necessarily involve Cauchy stress.

L493-496: The notion of crevasse \*initiation\* being governed by viscous processes seems muddled to me. Crevasse initiation is a fundamentally discontinuous phenomenon, no? A crevasse initiates when a discontinuity -the crevasse-forms in a medium formerly approximated as continuous. Consider reframing for clarity.

This point is addressed in the general comments.

L505-507: In my view, most people accept that bulk calving will be implemented differently from local-scale collapse events. Perhaps your manuscript would do best to focus more directly on the latter?

This point is addressed in the general comments.

**Figures**

Fig 1: "Surface Energy Balance station (pink triangle)" - Not seeing a pink triangle. Is it the one nearest the cavity on panel (a)? That one appears orange to me, the same as the other two triangle markers.

Yes, the pink one is the one nearest the cavity on panel (a). We have now changed its color to red to make it easier to distinguish.

Fig 6: Please refer back to where each equivalent stress is defined in Section 3. I was initially very confused about why MPS was the only plot that showed compressive stresses in the center of the roof, where we are quite certain the stress should be compressive. But in fact, this is not giving local effective stress but rather the value of the maximum principal stress, the Hayhurst stress, etc., which would be compared with the failure threshold. Right?

We have added references to the corresponding equations for each criterion. As stated in the caption, the plot shows the equivalent stresses for each criterion as defined in Eqs. (1)–(4), which indeed represent the stress metrics that should be compared with the failure threshold, depending on the chosen failure criterion.

Regarding the negative values shown for the MPS in the previous version of the manuscript, we would like to thank you for this question, as it allowed us to identify a small inconsistency. First, let us recall that, by definition, the maximum principal stress  $\sigma_{\rm I}$  is defined such that

$$\sigma_{\rm III} < \sigma_{\rm II} < \sigma_{\rm I}.$$
 (3)

Because the upper surface is a free surface, one of the three principal directions corresponds to the surface normal, and the associated principal stress is zero. Consequently, in the continuous domain, the maximum principal stress at the surface is systematically either positive or zero, but cannot be negative. The small negative values shown for the MPS in the previous version of Fig. 6 were actually numerical artefacts, mostly arising from the difficulty of defining the surface normal on a finite-element mesh of a not perfectly planar surface. This is confirmed by the fact that, when we recompute the normal stress at the surface, i.e.  $\mathbf{n} \cdot (\boldsymbol{\sigma} \cdot \mathbf{n})$ , during post-processing, we do not obtain strictly zero values (as should theoretically be the case on a traction-free boundary) but small residuals that correspond exactly to the values computed for the principal stress associated with the normal direction.

Note that this does not affect any of our analyses, since inequality (3) naturally implies that wherever  $\sigma_{\rm I} = 0$ , both  $\sigma_{\rm II}$  and  $\sigma_{\rm III}$  are negative. In other words, an area where the MPS is zero corresponds to a compressive regime. This point is now clearly explained in the manuscript, and we have replaced the artificial negative values by zero for the MPS case in Fig. 6.

Fig 7: "Blue dots indicate mean vertical velocities derived from stake measurements, computed between two measurement dates" - Which two measurement dates? What's the temporal resolution here?

Overall, the vertical velocities are averaged over periods of 4 to 8 days for the three drainage sequences, as personnel were continuously present in the field. Outside the drainage periods, field visits were less frequent, and the vertical velocities are therefore averaged over longer intervals, ranging from 9 days (summer 2011, before pumping) to 41 days (summer 2012, before pumping). We have added this precision in the figure caption.

**References**

- Behn, M. D., Goldsby, D. L., and Hirth, G.: The role of grain size evolution in the rheology of ice: implications for reconciling laboratory creep data and the Glen flow law, The Cryosphere, 15, 4589–4605, https://doi.org/10.5194/tc-15-4589-2021, 2021.
- Bons, P. D., Kleiner, T., Llorens, M.-G., Prior, D. J., Sachau, T., Weikusat, I., and Jansen, D.: Greenland Ice Sheet: Higher Nonlinearity of Ice Flow Significantly Reduces Estimated Basal Motion, Geophysical Research Letters, 45, 6542–6548, https://doi.org/10.1029/2018GL078356, 2018.
- Brondex, J., Gagliardini, O., Gillet-Chaulet, F., and Chekki, M.: Comparing the long-term fate of a snow cave and a rigid container buried at Dome C, Antarctica, Cold Regions Science and Technology, 180, 103 164, https://doi.org/10.1016/j.coldregions.2020.103164, 2020.
- Christmann, J., Müller, R., and Humbert, A.: On nonlinear strain theory for a viscoelastic material model and its implications for calving of ice shelves, Journal of Glaciology, 65, 212–224, https://doi.org/10.1017/jog.2018.107, 2019.
- Colgan, W., Rajaram, H., Abdalati, W., McCutchan, C., Mottram, R., Moussavi, M. S., and Grigsby, S.: Glacier crevasses: Observations, models, and mass balance implications, Reviews of Geophysics, 54, 119–161, https://doi.org/10.1002/2015RG000504, 2016.
- Cornford, S. L., Seroussi, H., Asay-Davis, X. S., Gudmundsson, G. H., Arthern, R., Borstad, C., Christmann, J., Dias dos Santos, T., Feldmann, J., Goldberg, D., Hoffman, M. J., Humbert, A., Kleiner, T., Leguy, G., Lipscomb, W. H., Merino, N., Durand, G., Morlighem, M., Pollard, D., Rückamp, M., Williams, C. R., and Yu, H.: Results of the third Marine Ice Sheet Model Intercomparison Project (MISMIP+), The Cryosphere, 14, 2283–2301, https://doi.org/10.5194/tc-14-2283-2020, 2020.
- Cuffey, K. and Kavanaugh, J.: How nonlinear is the creep deformation of polar ice? A new field assessment, Geology, 39, 1027–1030, https://doi.org/10.1130/G32259.1, 2011.
- Cuffey, K. M. and Paterson, W. S. B.: The physics of glaciers, Academic Press, 2010.
- Dempsey, J., Adamson, R., and Mulmule, S.: Scale effects on the in-situ tensile strength and fracture of ice. Part II: First-year sea ice at Resolute, N.W.T., International Journal of Fracture, 95, 347–366, https://doi.org/10.1023/A:1018650303385, 1999.
- Duddu, R. and Waisman, H.: A temperature dependent creep damage model for polycrystalline ice, Mechanics of Materials, 46, 23–41, https://doi.org/10.1016/j.mechmat.2011.11.007, 2012.
- Durand, G., Gagliardini, O., de Fleurian, B., Zwinger, T., and Le Meur, E.: Marine ice sheet dynamics: Hysteresis and neutral equilibrium, Journal of Geophysical Research: Earth Surface, 114, https://doi.org/10.1029/2008JF001170, 2009.
- Favier, L., Gagliardini, O., Durand, G., and Zwinger, T.: A three-dimensional full Stokes model of the grounding line dynamics: effect of a pinning point beneath the ice shelf, The Cryosphere, 6, 101–112, https://doi.org/10.5194/tc-6-101-2012, 2012.

- Gagliardini, O., Gillet-Chaulet, F., Durand, G., Vincent, C., and Duval, P.: Estimating the risk of glacier cavity collapse during artificial drainage: The case of Tête Rousse Glacier, Geophysical Research Letters, 38, https://doi.org/10.1029/2011GL047536, 2011.
- Gagliardini, O., Weiss, J., Duval, P., and Montagnat, M.: On Duddu and Waisman (2012, 2013) concerning continuum damage mechanics applied to crevassing and iceberg calving, Journal of Glaciology, 59, 797–798, https://doi.org/10.3189/2013JoG13J049, 2013.
- Garambois, S., Legchenko, A., Vincent, C., and Thibert, E.: Ground-penetrating radar and surface nuclear magnetic resonance monitoring of an englacial water-filled cavity in the polythermal glacier of Tête Rousse, Geophysics, 81, WA131–WA146, https://doi.org/10.1190/geo2015-0125.1, 2016.
- Getraer, B. and Morlighem, M.: Increasing the Glen-Nye Power-Law Exponent Accelerates Ice-Loss Projections for the Amundsen Sea Embayment, West Antarctica, Geophysical Research Letters, 52, e2024GL112516, https://doi.org/10.1029/2024GL112516, 2025.
- Gilbert, A., Vincent, C., Wagnon, P., Thibert, E., and Rabatel, A.: The influence of snow cover thickness on the thermal regime of Tête Rousse Glacier (Mont Blanc range, 3200 m a.s.l.): Consequences for outburst flood hazards and glacier response to climate change, Journal of Geophysical Research: Earth Surface, 117, https://doi.org/10.1029/2011JF002258, 2012.
- Gilbert, A., Gagliardini, O., Vincent, C., and Wagnon, P.: A 3-D thermal regime model suitable for cold accumulation zones of polythermal mountain glaciers, Journal of Geophysical Research: Earth Surface, 119, 1876–1893, https://doi.org/10.1002/2014JF003199, 2014.
- Gillet-Chaulet, F., Hindmarsh, R. C. A., Corr, H. F. J., King, E. C., and Jenkins, A.: In-situquantification of ice rheology and direct measurement of the Raymond Effect at Summit, Greenland using a phase-sensitive radar, Geophysical Research Letters, 38, https://doi.org/10.1029/2011GL049843, 2011.
- Grinsted, A., Rathmann, N. M., Mottram, R., Solgaard, A. M., Mathiesen, J., and Hvidberg, C. S.: Failure strength of glacier ice inferred from Greenland crevasses, The Cryosphere, 18, 1947–1957, https://doi.org/10.5194/tc-18-1947-2024, 2024.
- Gudmundsson, G. H.: Ice-stream response to ocean tides and the form of the basal sliding law, The Cryosphere, 5, 259–270, https://doi.org/10.5194/tc-5-259-2011, 2011.
- Huth, A., Duddu, R., and Smith, B.: A Generalized Interpolation Material Point Method for Shallow Ice Shelves. 2: Anisotropic Nonlocal Damage Mechanics and Rift Propagation, Journal of Advances in Modeling Earth Systems, 13, e2020MS002292, https://doi.org/10.1029/2020MS00229210.1002/essoar.10503748.1, 2021.
- Huth, A., Duddu, R., Smith, B., and Sergienko, O.: Simulating the processes controlling ice-shelf rift paths using damage mechanics, Journal of Glaciology, p. 1–14, https://doi.org/10.1017/jog.2023.71, 2023.
- Jiménez, S., Duddu, R., and Bassis, J.: An updated-Lagrangian damage mechanics formulation for modeling the creeping flow and fracture of ice sheets, Computer Methods in Applied Mechanics and Engineering, 313, 406–432, https://doi.org/10.1016/j.cma.2016.09.034, 2017.
- Jouvet, G., Picasso, M., Rappaz, J., Huss, M., and Funk, M.: Modelling and Numerical Simulation of the Dynamics of Glaciers Including Local Damage Effects, Math. Model. Nat. Phenom., 6, 263–280, https://doi.org/10.1051/mmnp/20116510, 2011.
- Krug, J., Weiss, J., Gagliardini, O., and Durand, G.: Combining damage and fracture mechanics to model calving, The Cryosphere, 8, 2101–2117, https://doi.org/10.5194/tc-8-2101-2014, 2014.
- Larour, E., Seroussi, H., Morlighem, M., and Rignot, E.: Continental scale, high order, high spatial resolution, ice sheet modeling using the Ice Sheet System Model (ISSM), Journal of Geophysical Research: Earth Surface, 117, https://doi.org/https://doi.org/10.1029/2011JF002140, 2012.
- Legchenko, A., Descloitres, M., Vincent, C., Guyard, H., Garambois, S., Chalikakis, K., and Ezersky, M.: Three-dimensional magnetic resonance imaging for groundwater, New Journal of Physics, 13, 025022, https://doi.org/10.1088/1367-2630/13/2/025022, 2011.

- Legchenko, A., Vincent, C., Baltassat, J. M., Girard, J. F., Thibert, E., Gagliardini, O., Descloitres, M., Gilbert, A., Garambois, S., Chevalier, A., and Guyard, H.: Monitoring water accumulation in a glacier using magnetic resonance imaging, The Cryosphere, 8, 155–166, https://doi.org/10.5194/tc-8-155-2014, 2014.
- MacAyeal, D. R., Sergienko, O. V., and Banwell, A. F.: A model of viscoelastic ice-shelf flexure, Journal of Glaciology, 61, 635–645, https://doi.org/10.3189/2015JoG14J169, 2015.
- Mercenier, R., Lüthi, M. P., and Vieli, A.: Calving relation for tidewater glaciers based on detailed stress field analysis, The Cryosphere, 12, 721–739, https://doi.org/10.5194/tc-12-721-2018, 2018.
- Mercenier, R., Lüthi, M. P., and Vieli, A.: A Transient Coupled Ice Flow-Damage Model to Simulate Iceberg Calving From Tidewater Outlet Glaciers, Journal of Advances in Modeling Earth Systems, 11, 3057–3072, https://doi.org/10.1029/2018MS001567, 2019.
- Millstein, J. D., Minchew, B. M., and Pegler, S. S.: Ice viscosity is more sensitive to stress than commonly assumed, Communications Earth and Environment, 3, 57, https://doi.org/10.1038/s43247-022-00385-x10.31223/x5d32x, 2022.
- Mobasher, M. E., Duddu, R., Bassis, J. N., and Waisman, H.: Modeling hydraulic fracture of glaciers using continuum damage mechanics, Journal of Glaciology, 62, 794–804, https://doi.org/10.1017/jog.2016.68, 2016.
- Paterson, W. S. B. and Savage, J. C.: Measurements on Athabasca Glacier relating to the flow law of ice, Journal of Geophysical Research (1896-1977), 68, 4537–4543, https://doi.org/10.1029/JZ068i015p04537, 1963.
- Pattyn, F., Perichon, L., Aschwanden, A., Breuer, B., de Smedt, B., Gagliardini, O., Gudmundsson, G. H., Hindmarsh, R. C. A., Hubbard, A., Johnson, J. V., Kleiner, T., Konovalov, Y., Martin, C., Payne, A. J., Pollard, D., Price, S., Rückamp, M., Saito, F., Souček, O., Sugiyama, S., and Zwinger, T.: Benchmark experiments for higher-order and full-Stokes ice sheet models (ISMIP-HOM), The Cryosphere, 2, 95–108, https://doi.org/10.5194/tc-2-95-2008, 2008.
- Petrovic, J.: Review mechanical properties of ice and snow, Journal of materials science, 38, 1–6, https://doi.org/10.1023/A:1021134128038, 2003.
- Pralong, A. and Funk, M.: Dynamic damage model of crevasse opening and application to glacier calving, Journal of Geophysical Research: Solid Earth, 110, 1–12, https://doi.org/10.1029/2004JB003104, 2005.
- Ranganathan, M. and Minchew, B.: A modified viscous flow law for natural glacier ice: Scaling from laboratories to ice sheets, Proceedings of the National Academy of Sciences, 121, e2309788121, https://doi.org/10.1073/pnas.2309788121, 2024.
- Raymond, C. F.: Inversion of flow Measurements for Stress and Rheological Parameters in a Valley Glacier, Journal of Glaciology, 12, 19–44, https://doi.org/10.3189/S0022143000022681, 1973.
- Rosier, S. H. R., Gudmundsson, G. H., and Green, J. A. M.: Insights into ice stream dynamics through modelling their response to tidal forcing, The Cryosphere, 8, 1763–1775, https://doi.org/10.5194/tc-8-1763-2014, 2014.
- Sun, S. and Gudmundsson, G. H.: The speedup of Pine Island Ice Shelf between 2017 and 2020: revaluating the importance of ice damage, Journal of Glaciology, 69, 1983–1991, https://doi.org/10.1017/jog.2023.76, 2023.
- Surawy-Stepney, T., Hogg, A. E., Cornford, S. L., and Davison, B. J.: Episodic dynamic change linked to damage on the Thwaites Glacier Ice Tongue, Nature Geoscience, 16, 37–43, https://doi.org/10.1038/s41561-022-01097-9, 2023.
- Sykes, H. J., Murray, T., and Luckman, A.: The location of the grounding zone of Evans Ice Stream, Antarctica, investigated using SAR interferometry and modelling, Annals of Glaciology, 50, 35–40, https://doi.org/10.3189/172756409789624292, 2009.
- Thomas, R. H., MacAyeal, D. R., Bentley, C. R., and Clapp, J. L.: The Creep of Ice, Geothermal Heat Flow, and Roosevelt Island, Antarctica, Journal of Glaciology, 25, 47–60, https://doi.org/10.3189/S0022143000010273, 1980.

- Ultee, L., Meyer, C., and Minchew, B.: Tensile strength of glacial ice deduced from observations of the 2015 eastern Skaftá cauldron collapse, Vatnajökull ice cap, Iceland, Journal of Glaciology, 66, 1024–1033, https://doi.org/10.1017/jog.2020.65, 2020.
- Vaughan, D. G.: Relating the occurrence of crevasses to surface strain rates, Journal of Glaciology, 39, 255–266, https://doi.org/10.3189/S0022143000015926, 1993.
- Vaughan, D. G.: Tidal flexure at ice shelf margins, Journal of Geophysical Research: Solid Earth, 100, 6213–6224, https://doi.org/10.1029/94JB02467, 1995.
- Vincent, C., Garambois, S., Thibert, E., Lefèbvre, E., Le Meur, E., and Six, D.: Origin of the outburst flood from Glacier de Tête Rousse in 1892 (Mont Blanc area, France), Journal of Glaciology, 56, 688–698, https://doi.org/10.3189/002214310793146188, 2010.
- Vincent, C., Descloitres, M., Garambois, S., Legchenko, A., Guyard, H., and Gilbert, A.: Detection of a subglacial lake in Glacier de Tête Rousse (Mont Blanc area, France), Journal of Glaciology, 58, 866–878, https://doi.org/10.3189/2012JoG11J179, 2012.
- Vincent, C., Thibert, E., Gagliardini, O., Legchenko, A., Gilbert, A., Garambois, S., Condom, T., Baltassat, J., and Girard, J.: Mechanisms of subglacial cavity filling in Glacier de Tête Rousse, French Alps, Journal of Glaciology, 61, 609–623, https://doi.org/10.3189/2015JoG14J238, 2015.
- Zeitz, M., Levermann, A., and Winkelmann, R.: Sensitivity of ice loss to uncertainty in flow law parameters in an idealized one-dimensional geometry, The Cryosphere, 14, 3537–3550, https://doi.org/10.5194/tc-14-3537-2020, 2020.